# Fibrillar Aβ triggers microglial proteome alterations and dysfunction in Alzheimer mouse models

Laura Sebastian Monasor[1,2†], Stephan A Müller[1†], Alessio Vittorio Colombo[1], Gaye Tanrioever[3,4], Jasmin König[1,5], Stefan Roth[6], Arthur Liesz[6,7], Anna Berghofer[8], Anke Piechotta[9], Matthias Prestel[6], Takashi Saito[10,11], Takaomi C Saido[10], Jochen Herms[1,7,12], Michael Willem[13], Christian Haass[1,7,13], Stefan F Lichtenthaler[1,7,8*], Sabina Tahirovic[1*]

[1]German Center for Neurodegenerative Diseases (DZNE), Munich, Germany; [2]Graduate School of Systemic Neuroscience, Ludwig-Maximilians-University, Munich, Germany; [3]German Center for Neurodegenerative Diseases (DZNE), Tübingen, Germany; [4]Department of Cellular Neurology, Hertie-Institute for Clinical Brain Research, University of Tübingen, Tübingen, Germany; [5]Faculty of Chemistry, Technical University of Munich, Garching, Germany; [6]Institute for Stroke and Dementia Research (ISD), University Hospital, LMU, Munich, Germany; [7]Munich Cluster for Systems Neurology (SyNergy), Munich, Germany; [8]Neuroproteomics, School of Medicine, Klinikum Rechts der Isar, Technical University, Munich, Germany; [9]Department of Molecular Drug Design and Target Validation, Fraunhofer Institute for Cell Therapy and Immunology, Halle, Germany; [10]Laboratory for Proteolytic Neuroscience, RIKEN Center for Brain Science Institute, Wako, Japan; [11]Department of Neurocognitive Science, Nagoya City University Graduate School of Medical Science, Nagoya, Japan; [12]Center for Neuropathology and Prion Research, Ludwig-Maximilians-Universität München, Munich, Germany; [13]Biomedical Center (BMC), Ludwig-Maximilians Universität München, Munich, Germany

*For correspondence:
stefan.lichtenthaler@dzne.de (SFL);
sabina.tahirovic@dzne.de (ST)

[†]These authors contributed equally to this work

Competing interests: The authors declare that no competing interests exist.

**Abstract** Microglial dysfunction is a key pathological feature of Alzheimer's disease (AD), but little is known about proteome-wide changes in microglia during the course of AD and their functional consequences. Here, we performed an in-depth and time-resolved proteomic characterization of microglia in two mouse models of amyloid β (Aβ) pathology, the overexpression APPPS1 and the knock-in APP-NL-G-F (APP-KI) model. We identified a large panel of Microglial Aβ Response Proteins (MARPs) that reflect heterogeneity of microglial alterations during early, middle and advanced stages of Aβ deposition and occur earlier in the APPPS1 mice. Strikingly, the kinetic differences in proteomic profiles correlated with the presence of fibrillar Aβ, rather than dystrophic neurites, suggesting that fibrillar Aβ may trigger the AD-associated microglial phenotype and the observed functional decline. The identified microglial proteomic fingerprints of AD provide a valuable resource for functional studies of novel molecular targets and potential biomarkers for monitoring AD progression or therapeutic efficacy.

## Introduction

Microglia play fundamental roles in a variety of neurodegenerative diseases, including AD (*McQuade and Blurton-Jones, 2019*). Changes in brain immunity, together with extracellular Aβ deposition and neurofibrillary tangles, are major pathological culprits in AD (*Gjoneska et al., 2015*;

**eLife digest** Alzheimer's disease is a progressive, irreversible brain disorder. Patients with Alzheimer's have problems with memory and other mental skills, which lead to more severe cognitive decline and, eventually, premature death. This is due to increasing numbers of nerve cells in the brain dying over time. A distinctive feature of Alzheimer's is the abnormally high accumulation of a protein called amyloid-β, which forms distinctive clumps in the brain termed 'plaques'.

The brain has a type of cells called the microglia that identify infections, toxic material and damaged cells, and prevent these from building up by clearing them away. In Alzheimer's disease, however, the microglia do not work properly, which is thought to contribute to the accumulation of amyloid-β plaques. This means that people with mutations in the genes important for the microglia activity are also at higher risk of developing the disease.

Although problems with the microglia play an important role in Alzheimer's, researchers still do not fully understand why microglia stop working in the first place. It is also not known exactly when and how the microglia change as Alzheimer's disease progresses. To unravel this mystery, Sebastian Monasor, Müller et al. carried out a detailed study of the molecular 'fingerprints' of microglia at each key stage of Alzheimer's disease.

The experiments used microglia cells from two different strains of genetically altered mice, both of which develop the hallmarks of Alzheimer's disease, including amyloid-β plaques, at similar rates. Analysis of the proteins in microglia cells from both strains revealed distinctive, large-scale changes corresponding to successive stages of the disease – reflecting the gradual accumulation of plaques. Obvious defects in microglia function also appeared soon after plaques started to build up.

Microscopy imaging of the brain tissue showed that although amyloid-β plaques appeared at the same time, they looked different in each mouse strain. In one, plaques were more compact, while in the other, plaques appeared 'fluffier', like cotton wool. In mice with more compacted plaques, microglia recognized the plaques earlier and stopped working sooner, suggesting that plaque structure and microglia defects could be linked.

These results shed new light on the role of microglia and their changing protein 'signals' during the different stages of Alzheimer's disease. In the future, this information could help identify people at risk for the disease, so that they can be treated as soon as possible, and to design new therapies to make microglia work again.

*Guillot-Sestier and Town, 2013*; *Holtzman et al., 2011*; *Shi and Holtzman, 2018*). The importance of microglia in AD pathogenesis is well illustrated by the increasing number of identified AD risk genes which are expressed in microglia and have functions in brain immunity (*Cuyvers and Sleegers, 2016*; *Guerreiro et al., 2013*; *Jansen et al., 2019*; *Jonsson et al., 2013*; *Karch and Goate, 2015*; *Lambert et al., 2009*; *Naj et al., 2011*; *Sims et al., 2017*). For example, the triggering receptor expressed on myeloid cells 2 (*TREM2*) and apolipoprotein E (*APOE*) are major genetic risk factors for sporadic AD that are expressed by plaque-associated microglia and involved in Aβ clearance (*Bradshaw et al., 2013*; *Castellano et al., 2011*; *Kleinberger et al., 2014*; *Parhizkar et al., 2019*; *Reddy et al., 2009*; *Wang et al., 2015*). It has also been shown that microglial phagocytosis decays over the course of AD (*Hickman et al., 2008*; *Koellhoffer et al., 2017*; *Orre et al., 2014a*; *Solito and Sastre, 2012*; *Zuroff et al., 2017*). Along these lines, Aβ clearance was found reduced in sporadic AD and it is assumed to be a key factor in the pathogenesis (*Mawuenyega et al., 2010*; *Saido, 1998*; *Wildsmith et al., 2013*). Importantly, Aβ clearance defects in AD microglia are reversible (*Daria et al., 2017*; *Krabbe et al., 2013*) and enhancing microglial phagocytic function has been explored as a therapeutic approach since substantial reduction of Aβ burden in mice appears to correlate with cognitive benefits (*Bacskai et al., 2001*; *Bard et al., 2000*; *Bohrmann et al., 2012*; *Janus et al., 2000*; *Lathuilière et al., 2016*; *Morgan et al., 2000*; *Nicoll et al., 2006*; *Nicoll et al., 2003*; *Schenk et al., 1999*; *Schlepckow et al., 2020*; *Sevigny et al., 2016*; *Wilcock et al., 2004*). However, when and how microglia change along AD progression is still not clear. Thus, understanding molecular alterations of microglia at different stages of AD is crucial and a pre-requisite for developing safe and efficacious therapy.

Transcriptional expression profiles of microglia were previously revealed under physiological, neurodegenerative or neuroinflammatory conditions (*Butovsky et al., 2014*; *Galatro et al., 2017*; *Gosselin et al., 2017*; *Götzl et al., 2019*; *Grabert et al., 2016*; *Holtman et al., 2015*; *Kamphuis et al., 2016*; *Krasemann et al., 2017*; *Mazaheri et al., 2017*; *Orre et al., 2014a*; *Orre et al., 2014b*; *Wang et al., 2015*; *Yin et al., 2017*). Transcriptional signatures were also recently reported at single-cell resolution, demonstrating regional and functional heterogeneity of brain myeloid cells (*Hammond et al., 2019*; *Jordão et al., 2019*; *Keren-Shaul et al., 2017*; *Mathys et al., 2017*; *Sala Frigerio et al., 2019*; *Zhou et al., 2020*). In neurodegenerative mouse models, two major profiles have been proposed along the spectrum of microglial alterations. One is the homeostatic microglial signature that occurs under physiological conditions and is characterized by the expression of several genes, including *P2ry12*, *Tmem119* and *Cx3cr1*. The other key signatures, referred to as disease-associated microglia (DAM), microglial neurodegenerative phenotype (MGnD) or activated response microglia (ARM) are observed under neurodegenerative conditions (*Keren-Shaul et al., 2017*; *Krasemann et al., 2017*; *Sala Frigerio et al., 2019*) and characterized by increased expression of *Apoe*, *Trem2*, *Cd68*, *Clec7a* and *Itgax* (*Cd11c),* among others. These changes were quantified using RNA transcripts, but transcript levels do not necessarily reflect protein levels which ultimately control cell function (*Böttcher et al., 2019*; *Mrdjen et al., 2018*; *Sharma et al., 2015*). Importantly, a recent study postulated that transcriptomic profiles of microglia from another AD mouse model (5xFAD) do not correlate well with proteomic changes (*Rangaraju et al., 2018*), suggesting the existence of additional translational or post-translational regulation mechanisms in AD microglia. Additionally, little is known about Aβ-associated changes in the microglial proteome in a time-resolved manner, or which proteome alterations underscore microglial dysfunction. Accordingly, we analyzed the microglial proteome at distinct stages of Aβ pathology in two commonly used mouse models of amyloidosis; the APPPS1 (*Radde et al., 2006*), and the APP-KI mice (*Saito et al., 2014*). In contrast to the APPPS1 mouse model that overexpresses mutated human amyloid precursor protein (APP) and presenilin-1 (PS1), the APP-KI model bears endogenous levels of APP with a humanized Aβ sequence containing three AD mutations (NL-G-F), and has no alterations of PS1 (*Radde et al., 2006*; *Saito et al., 2014*).

Our study determines the proteome of microglia from APPPS1 and APP-KI mice in a time resolved manner, starting from pre-deposition to early, middle and advanced stages of amyloid deposition and reveals a panel of Microglial Aβ Response Proteins (MARPs) that progressively change throughout Aβ accumulation. Although both mouse models display severe microglial alterations at late stages of Aβ pathology, the occurrence of MARP signatures differs and appears earlier in the APPPS1 mice. Strikingly, the kinetic differences in proteomic profiles correlated with the presence of fibrillar Aβ, rather than dystrophic neurites, suggesting that fibrillar Aβ aggregates may trigger the AD-associated microglial phenotype and corresponding functional decline. The time-resolved microglial profiles may serve as benchmark proteomic signatures for investigating novel microglial targets or monitoring the efficacy of future pre-clinical studies aiming at microglial repair.

## Results

### APPPS1 microglia develop AD-associated proteomic signatures earlier compared to APP-KI microglia

Amyloid plaque deposits appear at similar ages (between 6–8 weeks) in APPPS1 and APP-KI mouse models (*Radde et al., 2006*; *Saito et al., 2014*). To reveal the dynamics of microglial proteomic alterations across different amyloid stages, we analyzed microglia from 1, 3, 6 and 12 month old APPPS1 and APP-KI mice and their corresponding age-matched wild-type (WT) mice (*Figure 1—figure supplement 1A*). To facilitate proteomic analysis, we first optimized the microglial isolation procedure. CD11b positive microglia were isolated from mouse cerebrum using MACS technology. The purity of the CD11b-enriched fraction was controlled by fluorescence activated cell sorting (FACS), revealing that 97% of isolated cells were CD11b positive (*Figure 1—source data 1A*). Of note, only 0.49% of CD11b positive cells were detected in the CD11b-depleted fraction (*Figure 1—source data 1B*), demonstrating high isolation efficiency. Isolated microglia were lysed and then measured by LC-MS/MS using label-free quantification (LFQ) of proteins. Next, we optimized the data acquisition method for microglial proteome analysis. Recently, it was shown that Data Independent

Acquisition (DIA) for LFQ of proteins identifies and quantifies consistently more peptides and proteins across multiple samples, compared to Data Dependent Acquisition (DDA) (*Bruderer et al., 2015*). Thus, we first evaluated the performance of DDA *vs.* DIA using microglial lysates from WT and APPPS1 mice. DDA identified 53912 peptides on average compared to 74281 peptides identified by DIA, representing a 37.8% increase in detection by DIA method (*Supplementary file 1*). Overall, the main advantage of DIA was the improved consistency of protein quantifications among the replicates and the identification of proteins with lower abundance, leading to a 29% increase of relatively quantified proteins from 4412 with DDA to 5699 with DIA (*Figure 1—figure supplement 1B and C*, *Supplementary file 1*). Therefore, we chose the DIA acquisition method to also generate the proteome dataset of APP-KI microglia. We detected a consistent relative quantification of proteins with an overlap of 93.5% (5500 proteins) between the two investigated mouse models (*Figure 1—figure supplement 1D*), supporting our selection of DIA as a robust method for microglial proteomic analysis. For the comparative analysis of proteomic changes, we defined a threshold of a log2 fold change larger than 0.5 or smaller than −0.5 with a p-value less than 0.05, and significance after False Discovery Rate (FDR) correction. No data imputation was performed.

According to Aβ burden in both mouse models, we refer to one month of age as a pre-deposition stage, and to 3, 6 and 12 months of age as early, middle and advanced stages of amyloid pathology, respectively (*Figure 1—figure supplement 2*). At the pre-deposition stage (1 month), microglial proteomes of APPPS1 and APP-KI mice did not show significant alterations compared to WT (*Figure 1A and B*), demonstrating that microglia are not affected prior to development of Aβ pathology. At 3 months of age, microglia in APPPS1 mice already displayed a significant up-regulation of 332 proteins and down-regulation of 678 proteins, compared to WT microglia (*Figure 1C*, *Supplementary file 2A*). In contrast, APP-KI microglia were hardly affected at 3 months of age (*Figure 1D*, *Supplementary file 2B*), which is particularly surprising because both mouse models show comparable amyloid burden at this stage (*Figure 1—figure supplement 2*). At 6 months of age, microglia in APPPS1 mice displayed 309 up-regulated and 261 down-regulated proteins, compared to WT microglia (*Figure 1E*, *Supplementary file 2A*). In contrast to 3 months of age (*Figure 1D*), APP-KI mice displayed a substantial alteration of their microglial proteome at 6 months of age, illustrated by 140 up-regulated and 151 down-regulated proteins (*Figure 1F*, *Supplementary file 2B*). Still, microglial alterations in 6 month old APP-KI mice were less pronounced compared to the proteome of APPPS1 mice (*Figure 1E and F*). Noteworthy, by 12 months of age, APPPS1 microglia revealed a significant up-regulation of 776 proteins and down-regulation of 633 proteins, while APP-KI microglia displayed 704 up-regulated and 666 down-regulated proteins (*Figure 1G and H*, *Supplementary file 2A and B*). Overall, our data show that amyloid plaque accumulation triggers microglial progression towards an AD-associated phenotype in both mouse models, but that response dynamics are different in APPPS1 and APP-KI microglia.

## Identification of MARPs as signatures of early, middle and advanced amyloid stages

Next, we determined protein alterations that first appear in early, middle or advanced stages of Aβ deposition and remain altered through all analyzed stages, thus following amyloid accumulation. To this end, we selected the APPPS1 mouse model as a reference since it displays earlier changes and therefore provides a better time resolution of protein alterations to amyloid response, compared to the APP-KI model (*Figure 2A*). Only proteins with a consistent quantification in all samples of an age group were used for relative quantification. Furthermore, in order to determine robust and model-independent Aβ-triggered microglial alterations, we only selected MARPs that were altered with a significantly changed abundance in both mouse models (even if in APP-KI microglia changes appear later). This analysis identified 90 early, 176 middle, and 435 advanced MARPs (*Figure 2—source data 1*, *Figure 2—figure supplement 1A*). The most strongly regulated MARPs with early, middle and advanced response are displayed in corresponding heatmaps (*Figure 2B–D*).

Early MARPs included several of the previously identified transcriptional DAM markers (*Keren-Shaul et al., 2017*) such as ITGAX (CD11c), APOE, CLEC7a, LGALS3 (Galectin-3) and CD68, which were found with an increased abundance (*Figure 2B*). Moreover, proteins involved in antigen presentation such as CD74, H2-D1, TAP2, TAPBP and H2-K1 were revealed as up-regulated early MARPs. In addition, we discovered prominent changes in interferon signaling represented by the up-regulation of early MARPs, including MNDA, OAS1A, IFIT3, ISG15, GVIN1, STAT1 and 2

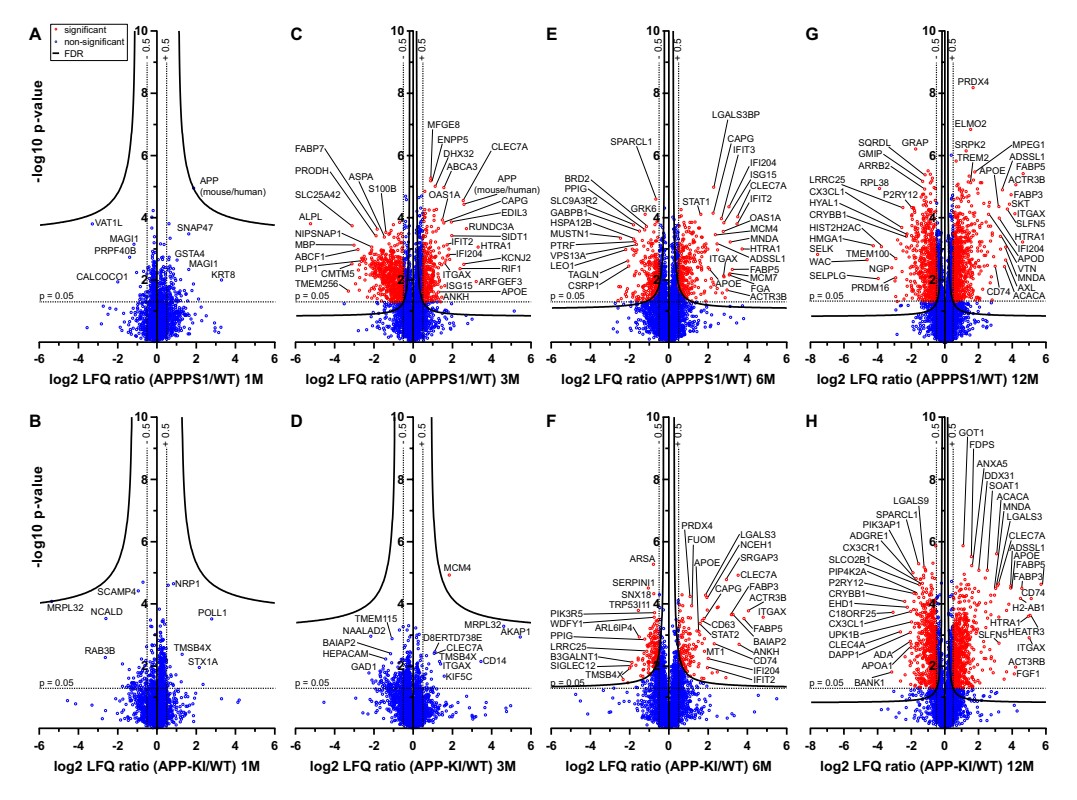

**Figure 1.** Quantitative proteomics of acutely isolated microglia. Volcano plots of APPPS1 and APP-KI *versus* WT microglia at 1 (**A and B**), 3 (**C and D**), 6 (**E and F**) and 12 (**G and H**) months of age. The minus log10 transformed p-value is plotted against the log2 transformed LFQ ratios (log2 fold changes). A permutation-based FDR estimation was applied which is visualized as hyperbolic curves. Proteins with a log2 LFQ ratio lower than −0.5 or higher than +0.5 with a p-value less than 0.05 which remain significantly changed after FDR correction (FDR = 0.05, $s_{0x00A0}$ = 0.1) are indicated as red circles. Non-significantly changed proteins are indicated as blue circles. Some selected proteins are marked with their gene names.

The online version of this article includes the following source data and figure supplement(s) for figure 1:

**Source data 1.** Quality control of microglial isolation using MACS.

**Figure supplement 1.** Improvement of the data acquisition method for quantitative proteomics of microglia.

**Figure supplement 2.** Aβ pathology in 3, 6 and 12 month old APPPS1 and APP-KI mice.

(*Figure 2B*). Even though early MARPs were mainly up-regulated, we also identified early MARPs with a decreased abundance, including KRAS, a protein involved in cell proliferation and the endocytosis regulator EHD2 among others (*Figure 2B*). A gene ontology (GO) cluster enrichment analysis of early MARPs revealed that up-regulated proteins were enriched for immune and viral response, interferon beta and cytokine response, antigen processing and presentation as well as biotic and lipid response (*Figure 3A*, *Figure 3—figure supplement 1A and D*). Thus, these processes represent first molecular alterations which progressively follow Aβ plaque pathology.

The middle MARPs included the up-regulated proteins FABP3, FABP5, CD63, TREM2, MIF and GUSB (*Figure 2C*), demonstrating a progressive conversion of the microglial proteome towards a disease state that accompanies Aβ accumulation. Importantly, middle MARPs also reveal down-regulation of the proposed homeostatic markers such as CX3CR1, TMEM119 and P2RY12 (*Figure 2C*). Among the down-regulated middle MARPs, we identified additional chemotaxis and cell migration related proteins like SYK, FER, CX3CL1, and BIN2 (*Figure 2C*, *Figure 2—source data 1*), underscoring a loss of key microglial functions throughout AD progression.

Advanced MARPs represent proteins that were only altered upon extensive amyloid pathology. This group included up-regulation of proteins involved in calcium ion binding such as NCAN, MYO5A, HPCAL4, TTYH1 and GCA and down-regulation of proteins that play a role in the endocytosis/lysosomal system such as TFEB, TFE3 and BIN1 (*Figure 2D*, *Figure 2—source data 1*). In addition, different G protein-coupled receptor signaling proteins, including GNG2, GNG5 and GNG10,

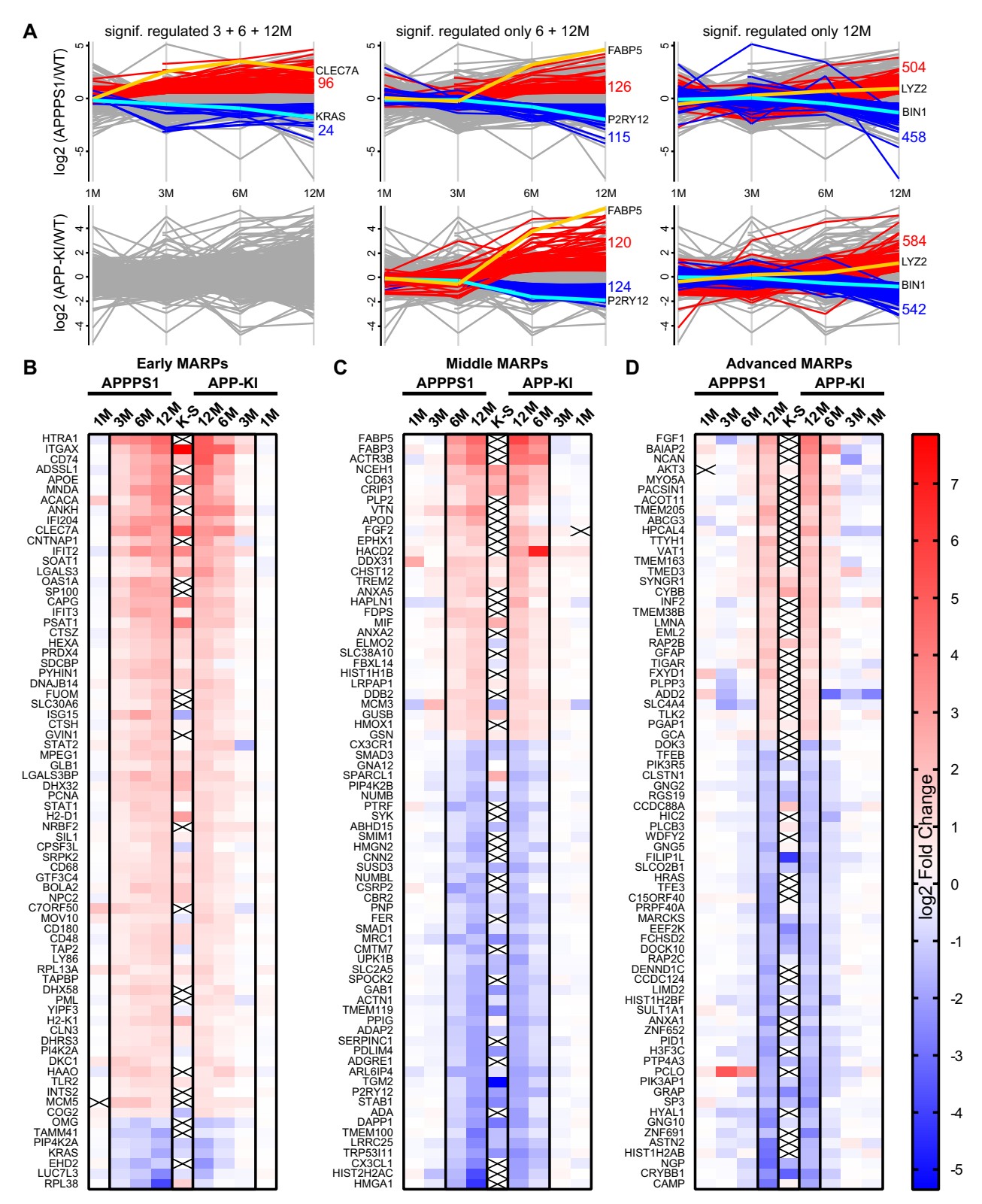

**Figure 2.** APPPS1 microglia display earlier proteomic changes compared to APP-KI microglia. (**A**) Profile plots of APPPS1 and APP-KI *versus* WT microglia at 1, 3, 6, and 12 months of age. Lines connect the average log2 fold changes of each protein at the different time points. Regulated proteins were grouped according to three profiles: significantly increased or decreased after FDR correction (log2 fold change >0.5 or<−0.5; p<0.05; FDR significant) at 3, 6, and 12 months or only at 6 and 12 or 12 months. Proteins that fulfill these criteria are indicated as red and blue lines for increased

*Figure 2 continued on next page*

*Figure 2 continued*

and decreased abundance, respectively. Selected up- or down-regulated proteins are indicated in orange and cyan. Heatmaps show the log2 fold changes of the top 74 up- or down-regulated proteins for early (B), middle (C) and advanced (D) MARPs and are compared to the log2 fold changes of transcripts from single cell transcriptome study (*Keren-Shaul et al., 2017*), indicated with K-S. Crosses indicate missing values.

The online version of this article includes the following source data and figure supplement(s) for figure 2:

**Source data 1.** Early, middle, and advanced MARPs.
**Figure supplement 1.** Comparison of the early, middle and advanced MARPs in APPPS1 and APP-KI mice.
**Figure supplement 2.** Relative quantification of proteins from seven AD risk genes in APPPS1.

---

also displayed a decreased abundance (*Figure 2D*). Furthermore, we observed a high correlation of MARP signatures between the two models at 12 months (*Figure 2—figure supplement 1B*).

A GO cluster enrichment analysis of middle and advanced MARPs identified down-regulation of biological processes including cell motility, migration and chemotaxis, as well as cell development and proliferation (*Figure 3B and C*, *Figure 3—figure supplement 1B,C,E and F*). Conversely, we found an up-regulation of protein glycosylation and carbohydrate metabolism (*Figure 3C*, *Figure 3— figure supplement 1E and F*). Additionally, alterations in ion transport processes involving ion homeostasis and pH regulation were also detected (*Figure 3C*). These findings indicate that after an initial inflammatory response, several cellular processes related to chemotaxis and phagocytosis are progressively dysregulated upon increased Aβ deposition.

Importantly, our proteomic analysis also detected alterations in proteins related to different genetic risk factors of AD (*Karch and Goate, 2015*), including significantly increased levels of APOE, TREM2, and INPP5D, and decreased levels of PLCG2, ABI3, and BIN1 in both mouse models (*Figure 2—figure supplement 2A and B*).

In addition, we compared MARP signatures with the previously published single cell transcriptome study of 5xFAD mice (*Keren-Shaul et al., 2017*) to visualize the overlap, as well as differences, between proteomic and transcriptomic microglial profiles (*Figure 2B–D*). When comparing the overlapping proteome of 12 months old APPPS1 and APP-KI mice with the transcriptome of 5xFAD mice (*Keren-Shaul et al., 2017*), we were able to quantify 3348 common proteins/transcripts, whereas 2152 and 2841 gene products were only quantified on protein and transcript level, respectively (*Figure 4A*). Comparison of our proteomic signatures with the microglial transcripts (*Keren-Shaul et al., 2017*) revealed an overlap of 227 unidirectionally regulated, whereas 263 and 849 gene products were only regulated on protein and transcript level, respectively (*Figure 4B*). While transcriptomics demonstrates a similar regulation of a large number of early MARPs, we found less overlap for middle and advanced MARPs (*Figure 2B–D*). We also identified proteins with an inverse regulation compared to transcriptomic signatures such as the early MARP RPL38, middle MARPs MCM3 and GFPT1 or advanced MARPs CDC88A, GALNT2, EIF4B and CHMP6 (*Figure 2B–D*, *Figure 2—source data 1*). Furthermore, the advanced MARP HEXB showed a consistent up-regulation in our proteomic analysis (*Figure 2—source data 1*), despite being previously anticipated as a homeostatic gene.

Overall, our study presents a robust and reliable method to track microglial proteome and provides a resource that maps changes in brain immunity during different phases of Aβ accumulation.

## Proteomic changes are detected in plaque-associated microglia

Next, we validated proteomic changes by western blot analysis using isolated microglia from 12 month old APPPS1 and APP-KI mice. This analysis confirmed the pronounced increase of the early MARPs APOE and CD68, the middle MARPs TREM2 and FABP5, as well as reduced levels of the advanced MARP CSF1R (*Figure 4C*) in both transgenic mouse models compared to WT mice. Furthermore, proteomic changes were also validated by immunohistochemistry in order to visualize spatial distribution of altered microglial proteins in APPPS1 and APP-KI mice. Immunohistological analysis of 3 month old APPPS1 mice already revealed increased immunoreactivity of selected MARPs such as CLEC7a (*Figure 5*), TREM2 (*Figure 5—figure supplement 1*) and APOE (*Figure 5— figure supplement 2*) that mark initial stages of microglial activation in AD. This increase was detected in IBA1 positive microglia that were clustering around amyloid plaques, but not in microglia further away from plaques and was – in agreement with our proteomic data – less pronounced

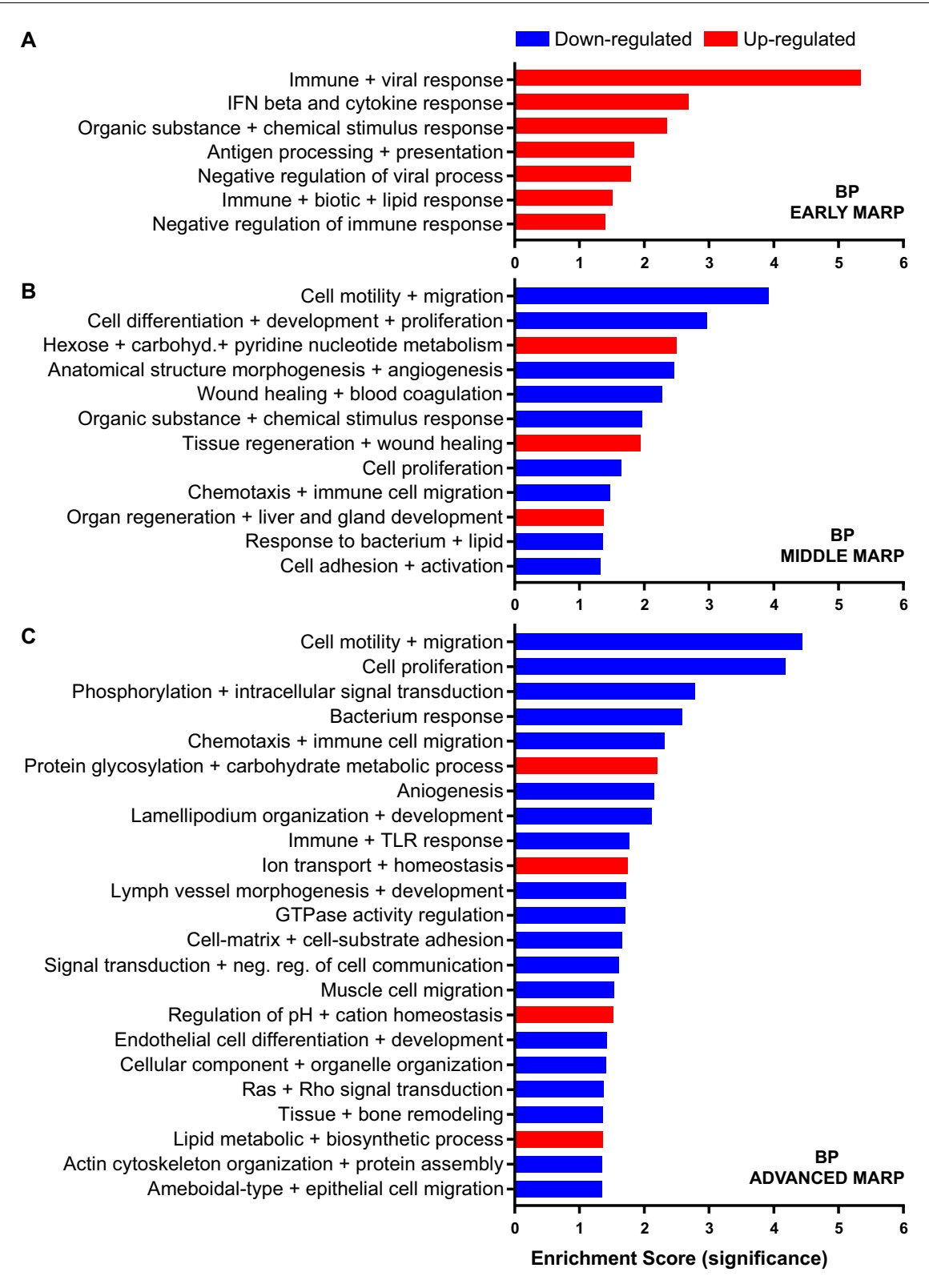

**Figure 3.** Gene ontology enrichment cluster analysis for biological process (BP) of MARPs. Bar graphs show the clustering of early MARPs (**A**), middle MARPs (**B**) and advanced MARPs (**C**) for BP. Up- and down-regulated proteins were analyzed separately using the web-based software tool DAVID 6.8 with all consistently quantified proteins (5500) as an individual background. Significantly enriched clusters (Enrichment Score >1.301) for up-and down-regulated proteins are indicated in red and blue, respectively.

*Figure 3 continued on next page*

*Figure 3 continued*

The online version of this article includes the following figure supplement(s) for figure 3:

**Figure supplement 1.** Gene ontology enrichment cluster analysis for cellular component (CC) and molecular function (MF) of MARPs.

in 3 month old APP-KI mice. Accordingly, at 12 months, both APPPS1 and APP-KI mice showed a similar increase in the levels of selected MARP CLEC7a (*Figure 6*) and decreased levels of TMEM119 (*Figure 7*) compared to the WT mice, once again in microglia surrounding amyloid plaques. Taken together, we validated selected microglial proteomic alterations from our dataset by applying biochemical and immunohistochemical methods. In addition, we confirmed the kinetic differences in AD-associated proteomic signatures of APPPS1 and APP-KI microglia. Our data suggest that

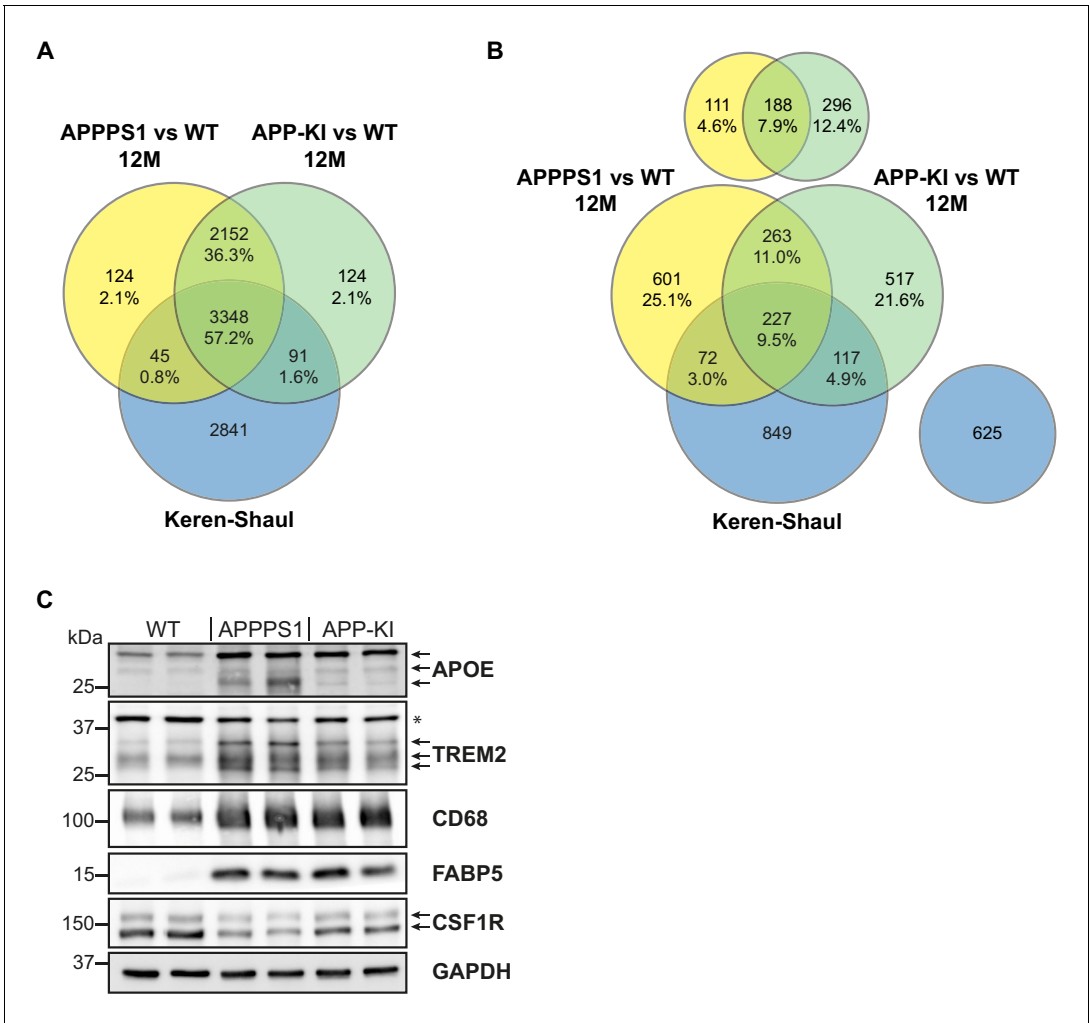

**Figure 4.** Comparison of quantified and regulated proteins of APPPS1 and APP-KI mice at 12 months with single cell transcriptomics and biochemical validation of regulated proteins. (**A**) The comparison of relatively quantified proteins of APPPS1 (yellow) and APP-KI (green) mice *versus* WT at 12 months shows a modest overlap with a single cell transcriptomics study (blue) (*Keren-Shaul et al., 2017*). An amount of 3348 proteins (57.2% of all quantified proteins) and their related transcripts were quantified in all three data sets. (**B**) The Venn diagram shows the overlap of significantly regulated proteins in APPPS1 (yellow) and APP-KI (green) mice *versus* WT at 12 months with unidirectionally regulated microglial transcripts (blue) (*Keren-Shaul et al., 2017*). Signatures which are specific either for the proteome or transcriptome dataset are separately depicted at the top or right, respectively. The indicated percentage values are based on all relatively quantified proteins. (**C**) Biochemical validation of proteomic data was performed using western blot analysis of microglial lysates from 12 month old mice. This analysis revealed an increased abundance of the up-regulated MARPs APOE, TREM2, CD68 and FABP5 as well as a decreased abundance of the down-regulated MARP CSF1R in APPPS1 and APP-KI compared to WT microglia. Arrows indicate antibody-detected specific bands and asterisk indicates unspecific bands.

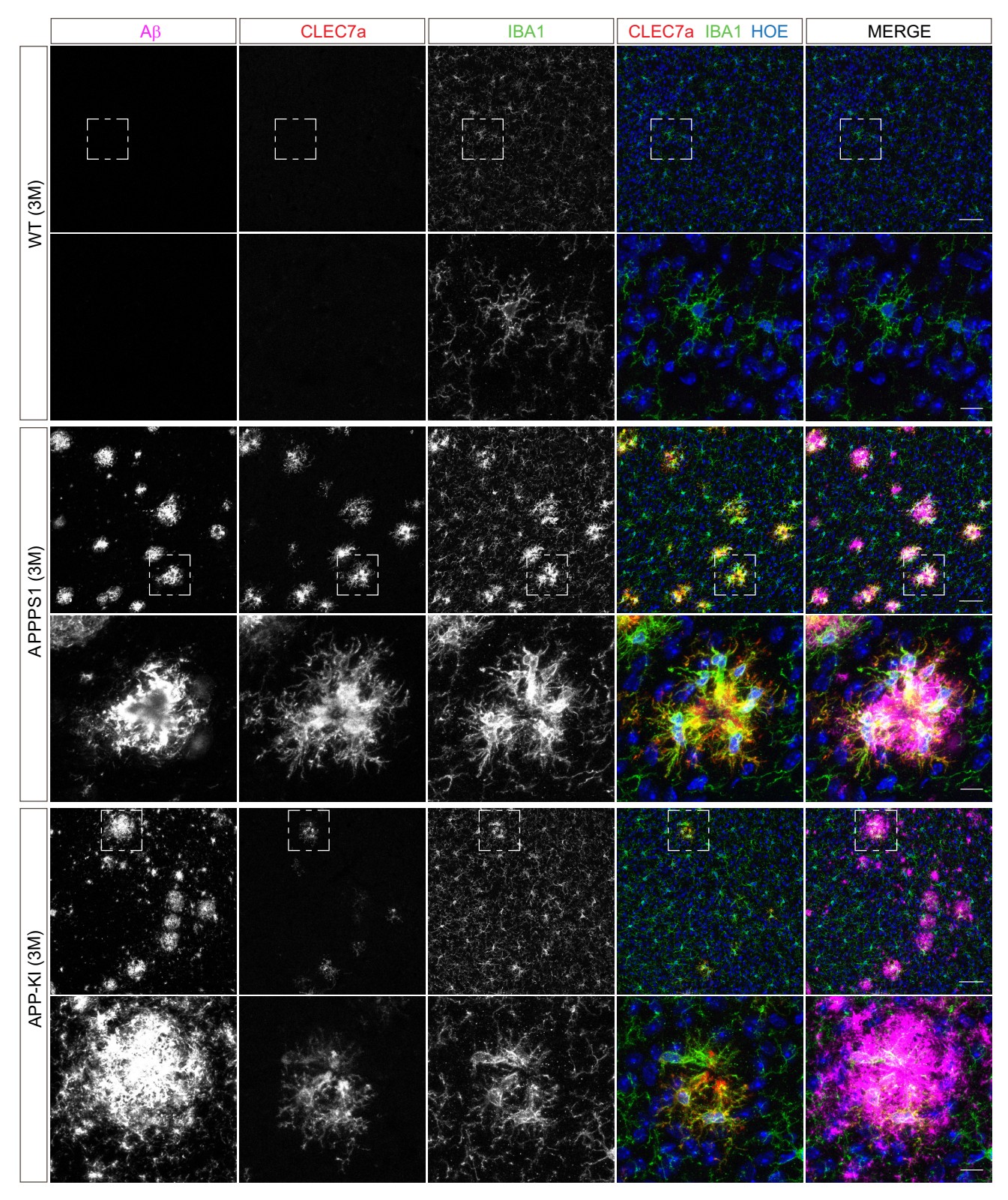

**Figure 5.** Microglial CLEC7a is prominently up-regulated in 3 month old APPPS1 mice. Immunohistochemical analysis of CLEC7a (red) reveals an increased abundance of this early MARP in IBA1 positive (green) APPPS1 microglia surrounding Aβ plaques (magenta) that is less prominent in APP-KI microglia. CLEC7a is barely detected in WT microglia. Hoechst (HOE, blue) was used for nuclear staining. Boxed regions in upper panels (scale bar: 50 μm) are shown with a higher magnification in lower panels (scale bar: 10 μm).

*Figure 5 continued on next page*

*Figure 5 continued*

The online version of this article includes the following figure supplement(s) for figure 5:

**Figure supplement 1.** Microglial TREM2 is prominently up-regulated in 3 month old APPPS1 mice.
**Figure supplement 2.** Microglial APOE is prominently up-regulated in 3 month old APPPS1 mice.

interaction between microglia and Aβ is likely triggering the proteomic changes as they could be observed in plaque-associated microglial population.

## APPPS1 and APP-KI mice show similar dynamics of amyloid plaque deposition, but differ in plaque fibrillization

The magnitude of proteomic microglial changes was found to correlate with Aβ plaque accumulation throughout disease progression. However, the appearance of MARP signatures differed between the models and occurred earlier in the APPPS1 mice (*Figure 1C and D*, *Figure 2A*) despite the comparable plaque load observed in both mouse models (*Figure 1—figure supplement 2*). Thus, it appears possible that the nature of amyloid plaques is different between the APPPS1 and APP-KI mice. To examine this, we analyzed amyloid plaques in 3, 6 and 12 month old APPPS1 and APP-KI mice by immunohistochemistry. We used the anti-Aβ antibody NAB228 (*Abner et al., 2018*) to detect Aβ plaques, and Thiazine Red (ThR) to visualize fibrillar amyloid (*Daria et al., 2017*; *Figure 8A*). In agreement with amyloid plaque pathology reported in this model (*Radde et al., 2006*), APPPS1 mice contained fibrillar amyloid plaque cores already at 3 months of age. In contrast, fibrillar Aβ was barely detectable in APP-KI mice at 3 months of age (*Figure 8A*). The amount of fibrillar Aβ in APP-KI mice increased at 6 and 12 months, but overall remained lower compared to the APPPS1 mice (*Figure 8A*). The reduced levels of fibrillar Aβ in APP-KI mice were also confirmed by biochemical analysis in which fibrillar Aβ was specifically detected via immunoblot of the insoluble brain fraction (*Figure 8B*). Despite the reduced levels of fibrillar Aβ, Aβ coverage was increased in 3 month old APP-KI compared to the APPPS1 mice (*Figure 8C*). To obtain further information on the conformational state of amyloid plaque cores in both mouse models, we performed spectral analysis using two luminescent conjugated oligothiophenes (LCOs) as reported previously (*Rasmussen et al., 2017*; *Figure 8—figure supplement 1A–D*). The quadro-formyl thiophene acetic acid (qFTAA) LCO binds to dense core amyloid fibrils and the hepta-formyl thiophene acetic acid (hFTAA) LCO seems to recognize both amyloid fibrils and less dense pre-fibrillar amyloid aggregates (*Klingstedt et al., 2011*; *Nyström et al., 2013*). In contrast to APP-KI, a prominent qFTAA signal was detected in APPPS1 mice at 3 months of age, revealing dense core fibrillar Aβ (*Figure 8—figure supplement 1A and C*). As expected, we could detect the hFTAA signal in both models at 3 months of age, visualizing the less dense, pre-fibrillar amyloid aggregates. To compare the levels of dense fibrillar *versus* pre-fibrillar Aβ, we quantified the ratio between peak intensities of qFTAA (emission at 502) and hFTAA (emission at 588) in both models at 3 and 12 months (*Figure 8—figure supplement 1D*). This analysis revealed significantly reduced levels of fibrillar Aβ in the APP-KI mice at the age of 3 months and only a trend towards reduction at the age of 12 months (*Figure 8—figure supplement 1A–D*). Thus, we demonstrate prominent differences in Aβ plaque fibrillization between APPPS1 and APP-KI mice at the age of 3 months.

## Microglial recruitment correlates with fibrillar Aβ

To determine what triggers microglial reactivity in AD, we first quantified microglial recruitment to Aβ plaques in both mouse models. This analysis was done at the early pathological stage (3 months), where we identified prominent differences in the proteome regulation (*Figure 1C and D*, *Figure 2A*) as well as in the amount of fibrillar Aβ (*Figure 8A and B*, *Figure 8—figure supplement 1A–C*) between the two mouse models. Immunohistochemical analysis revealed IBA1 positive, amoeboid microglia recruited to large, ThR positive, fibrillar Aβ aggregates in APPPS1 mice (*Figure 9A*). Of note, we observed intracellular fibrillar Aβ in APPPS1 microglia in close contact to the plaque core as previously reported (*Bolmont et al., 2008*). Despite the significantly smaller fibrillar Aβ aggregates in APP-KI mice, we could observe IBA1 positive microglia polarized towards the fibrillar Aβ, rather than to the surrounding non-fibrillar Aβ positive material (*Figure 9A*). Quantification analysis revealed increased clustering of IBA1 positive microglia around Aβ plaques in APPPS1 compared to

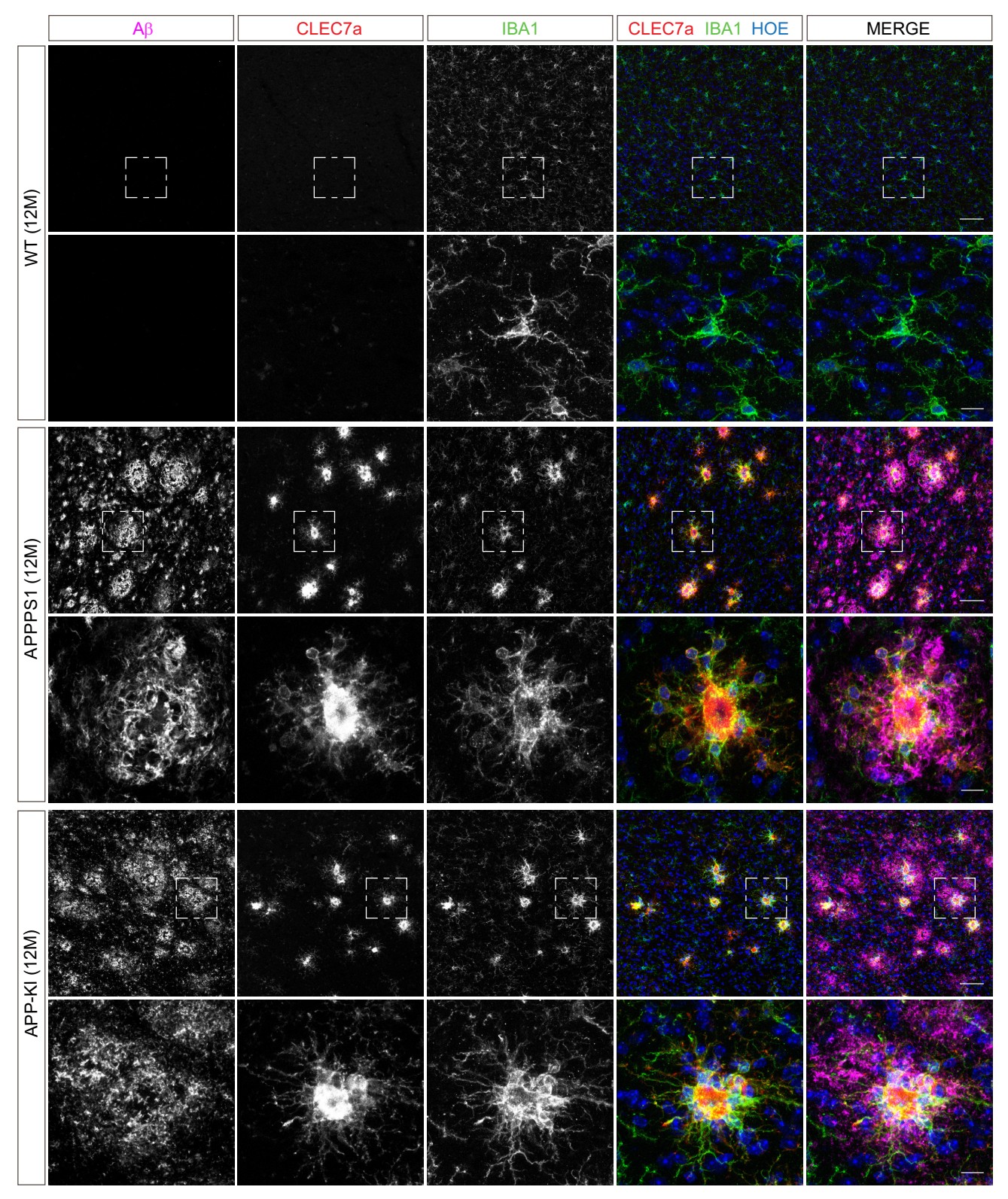

**Figure 6.** Microglial CLEC7a is increased in both AD mouse models at 12 months of age. Immunohistochemical analysis shows an up-regulation of CLEC7a (red) in IBA1 positive (green) APPPS1 and APP-KI microglia surrounding Aβ plaques (magenta), compared to WT where CLEC7a is barely detected. Hoechst (HOE, blue) was used for nuclear staining. Boxed regions in upper panels (scale bar: 50 µm) are shown with a higher magnification in lower panels (scale bar: 10 µm).

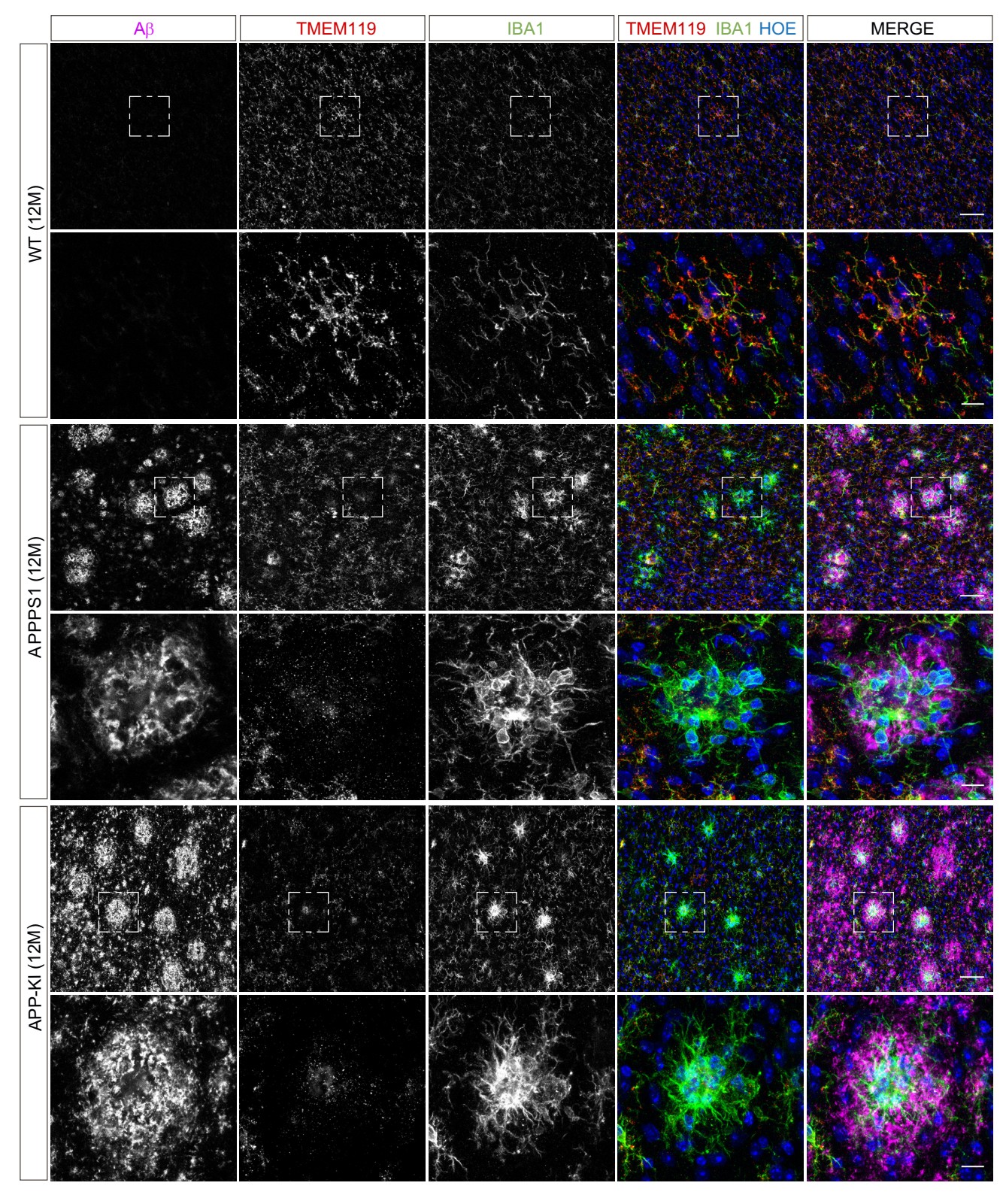

**Figure 7.** Microglial TMEM119 is down-regulated in both AD mouse models at 12 months of age. Immunohistochemical analysis of TMEM119 (red) shows a broad coverage signal of this homeostatic marker in IBA1 positive (green) WT microglia. In the APPPS1 and APP-KI mice, TMEM119 signal is reduced in IBA1 positive microglia surrounding Aβ plaques (magenta). Of note, increased TMEM119 signal was often found in the core of amyloid

*Figure 7 continued on next page*

*Figure 7 continued*

plaques in APPPS1 and APP-KI mice. Hoechst (HOE, blue) was used for nuclear staining. Boxed regions in upper panels (scale bar: 50 µm) are shown with a higher magnification in lower panels (scale bar: 10 µm) and illustrate reduced levels of TMEM119 in plaque associated microglia.

the APP-KI mice (*Figure 9B*), despite their overall larger Aβ plaque size (*Figure 9C*), supporting that fibrillar Aβ conformation, rather than plaque size, are responsible for microglial recruitment. Likewise, we observed increased CD68 immunoreactivity around Aβ plaques in the APPPS1 compared to the APP-KI mice (*Figure 9D and E*). However, CD68 signal per individual microglial cell in the plaque vicinity was similar in both models (*Figure 9F*), suggesting that differences in AD-associated microglial proteins are due to the number of recruited microglia rather than differences in their individual CD68 protein levels.

One of the Aβ modifications that can be readily detected in AD brains and favours fibrillar conformation is pyroglutamate-modified Aβ (pE3-Aβ) (*Dammers et al., 2015a*; *Dammers et al., 2015b*). To test whether pE3-Aβ may be triggering microglial reactivity, we assessed pE3-Aβ levels in 3 and 12 month old APPPS1 and APP-KI mice by immunohistochemistry, using a previously validated antibody (*Hartlage-Rübsamen et al., 2018*; *Figure 9—figure supplement 1A–D*). As we could only detect pE3-Aβ reactivity in 12 month old animals, but not at 3 months (*Figure 9—figure supplement 1A and B*), it is less likely that this modification is responsible for microglial recruitment and differences in the proteome between the models. Similarly as observed for fibrillar Aβ, levels of pE3-Aβ were reduced in the APP-KI compared to the APPPS1 mice (*Figure 9—figure supplement 1C and D*).

Besides Aβ, microglial recruitment has also been associated with neuritic damage (dystrophic neurites) (*Hemonnot et al., 2019*). Accordingly, we analyzed dystrophic neurite pathology in 3 month old APPPS1 and APP-KI mice, using an antibody against APP that accumulates in these structures (*Cummings et al., 1992*; *Sadleir et al., 2016*). As previously reported (*Radde et al., 2006*), amyloid plaques in the APPPS1 mice were surrounded by prominent dystrophic neurites (*Figure 9G*). Interestingly, despite the reduced load of fibrillar Aβ, we readily detected dystrophic neurites in the APP-KI mice (*Figure 9G*). Moreover, our quantification analysis revealed an increased dystrophic neurite area in the APP-KI compared to the APPPS1 mice (*Figure 9H*) and reduced number of microglial cells recruited to this area (*Figure 9I*). Therefore, the differences in early microglial recruitment to APPPS1 plaques and the consecutive proteomic changes are less likely to be triggered by dystrophic neurites.

Altogether, we hypothesize that microglial recruitment may be triggered by the fibrillar Aβ content of amyloid plaques, which drives the acquisition of MARP signatures.

## Phagocytic impairments correlate with the occurrence of MARP signatures

The differences observed in the dynamics of microglial response to amyloid in the APPPS1 and the APP-KI mice prompted us to examine the association between microglial phagocytic function and the appearance of MARP signatures. To this end, we assessed the phagocytic capacity of microglia from 3 and 6 month old APPPS1 and APP-KI mice compared to the corresponding age-matched WT microglia using the *E. coli*-pHrodo uptake assay (*Götzl et al., 2019*; *Kleinberger et al., 2014*). We already detected phagocytic impairments in 3 month old APPPS1 microglia, which was reflected by a prominent decrease in the amount of intracellular *E. coli* particles (*Figure 9J*, *Figure 9—source data 1A*) and a reduced number of CD11b positive cells that were capable of *E. coli* uptake (*Figure 9K*, *Figure 9—source data 1B*). Notably, APPPS1 phagocytic impairment did not change further in 6 month old microglia, suggesting that microglial functional deficits, as measured by the *E. coli* uptake assay, were fully established already at 3 months of age and characterized by early MARPs. In contrast, APP-KI microglia remained functional at 3 months, but at 6 months displayed similar impairments as seen in APPPS1 microglia (*Figure 9J and K*). Overall, we observed different kinetics of microglial dysfunction among mouse models which correlate with the appearance of MARPs and, in turn, with the presence of fibrillar Aβ.

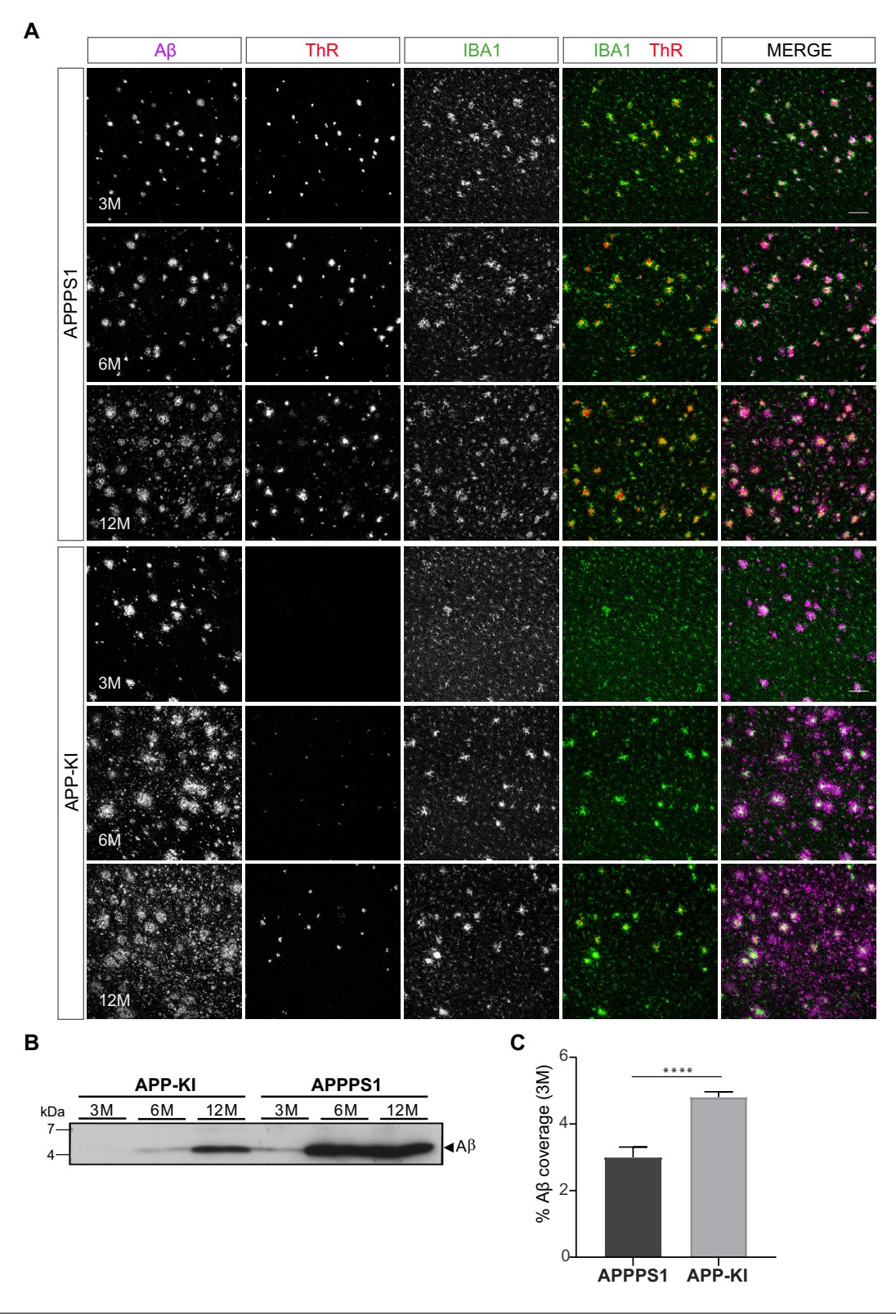

**Figure 8.** APPPS1 Aβ plaques display a higher content of fibrillar Aβ compared to the APP-KI plaques. (**A**) Immunohistochemical analysis showing total Aβ (magenta), fibrillar Aβ (ThR, red) and microglia (IBA1, green) in both mouse models at 3, 6 and 12 months of age. Scale bar: 100 μm **B**). Western blot analysis of insoluble Aβ at 3, 6 and 12 months of age confirms increased levels of fibrillar Aβ in APPPS1 compared to the APP-KI mice. (**C**).
*Figure 8 continued on next page*

*Figure 8 continued*

Quantification reveals increased total Aβ coverage in 3 month old APP-KI compared to the APPPS1 mice. Values are expressed as percentages of area covered by total Aβ and calculated as the mean (± SD) from N = 4 mice (****p<0.0001, unpaired two-tailed Student's T-test).

The online version of this article includes the following figure supplement(s) for figure 8:

**Figure supplement 1.** Amyloid plaque spectral characteristics show differences between 3 month old APPPS1 and APP-KI mice.

## Discussion

This study presents an in-depth and time-resolved proteome of microglia isolated across different stages of Aβ accumulation in the APPPS1 and APP-KI mouse models, resulting in the identification of early, middle, and advanced MARPs. We propose that the structure of amyloid plaques (fibrillar *versus* non-fibrillar) triggers the molecular alterations of microglia. Key microglial signatures encompass proteins with a central function in microglial biology and AD pathogenesis. Moreover, our functional analysis shows that early MARP signatures already reflect microglial phagocytic dysfunction.

To achieve robust and reproducible relative quantification of microglial proteins from single mice, we improved the yield of acutely isolated microglia to an average of $2 \times 10^6$ cells per mouse brain, compared to recently published protocols (*Flowers et al., 2017*; *Rangaraju et al., 2018*). Next, by establishing the more sensitive DIA method for protein quantification, we improved the number of consistently identified proteins by 29.3% and obtained an average of 5699 (APPPS1) and 5698 (APP-KI) relatively quantified proteins. Notably, our analysis enhanced the detection of low abundance proteins and did not require data imputation. Although we included both male and female mice for the analysis of microglial proteome, our study was not powered to detect sex-specific differences that have been reported in microglia (*Sala Frigerio et al., 2019*). The advancement to previous studies (*Rangaraju et al., 2018*; *Sharma et al., 2015*) is also exemplified by quantification of membrane proteins, including well known microglial homeostatic markers TMEM119 or P2RY12. We also measured alterations in proteins that were postulated to be only altered at the transcriptional level in AD microglia (*Rangaraju et al., 2018*), including up-regulation of middle MARPs FABP3, FABP5, PLP2 and MIF. In summary, our study achieved a major improvement in quantitative proteomic analysis of rodent microglia (*Flowers et al., 2017*; *Rangaraju et al., 2018*; *Thygesen et al., 2018*). This methodological advance enabled us to map microglial changes across diverse stages of Aβ pathology in two widely explored pre-clinical models of amyloidosis. Generated proteomic profiles characterize microglia under diseased conditions and can be used as a resource to track changes upon microglial therapeutic modification, such as Aβ immunotherapy. Such studies would facilitate discovery of clinically relevant molecular alterations that are necessary for microglial functional repair, monitoring disease progression and therapeutic efficacy.

The TREM2/APOE axis plays a key role in the regulation of the microglial transcriptional program and guides the homeostatic/DAM signature switch (*Jay et al., 2017*; *Keren-Shaul et al., 2017*; *Krasemann et al., 2017*). Our time-resolved proteomic analysis observed major rearrangements of the microglial proteomic landscape in both APPPS1 and APP-KI mice and revealed a partial overlap between MARPs and transcriptional profiles of DAM and homeostatic microglia (*Keren-Shaul et al., 2017*), but also identified additional microglial marker proteins throughout different stages of Aβ deposition.

Early MARPs include proteins of the interferon response, which is consistent with the recently identified interferon-responsive microglial sub-population in AD mice (*Sala Frigerio et al., 2019*). Numerous up-regulated early MARPs, including CD74, CTSD, CTSH, CTSZ, HEXA, GLB1, CD68, NPC2 and CLN3, reflect alterations in endo-lysosomal homeostasis as an early pathological insult in AD microglia (*Van Acker et al., 2019*). Additionally, factors of the fatty acid and cholesterol metabolism are altered throughout all pathological phases. Up-regulated are the early (APOE, ACACA, and SOAT1) middle (FABP3, FABP5, NCEH1, APOD, AACS, ACOX3, HACD2) and advanced MARPs (ACOT11, ACSBG1, ECHS1, ELOVL1, and FASN) and down-regulated are several middle and advanced MARPs (NAAA, FAM213B, HPGD, HPGDS, and PRKAB1), linking microglial lipid dyshomeostasis and AD pathology.

An inflammatory response in AD is suggested by the significant up-regulation of early MARPs LGALS3 and its binding protein (LGALS3BP). Recent findings indicated that the LGALS3/TREM2 signalling pathway, that acts as an inflammatory regulator of amyloid plaque formation, may also be of

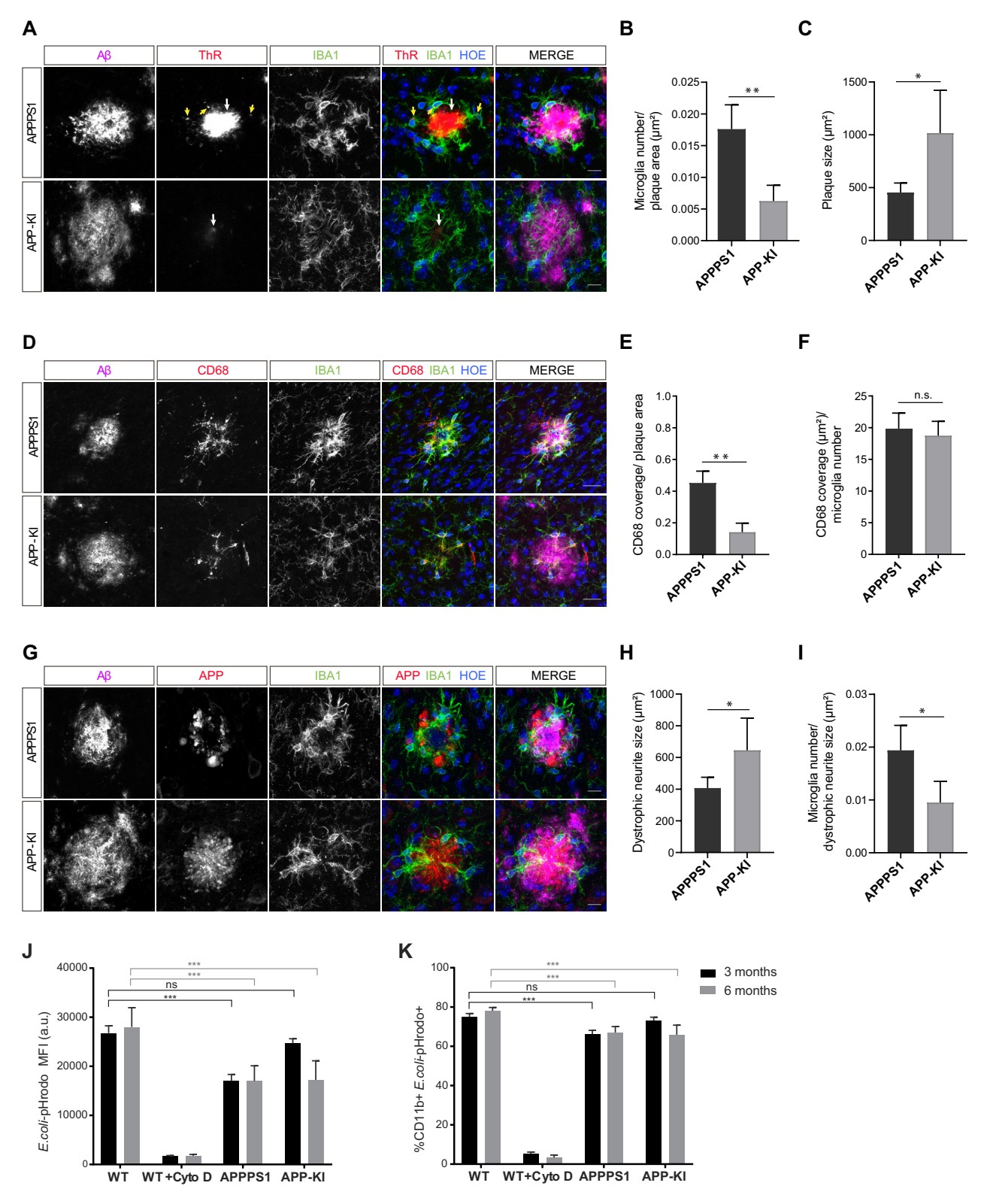

**Figure 9.** Fibrillar Aβ triggers higher microglial recruitment in 3 month old APPPS1 mice and correlates with a phagocytic impairment. (**A**) IBA1 positive (green) microglia polarize towards the fibrillar Aβ core (ThR, red, white arrow), rather than to the surrounding plaque halo (magenta) in 3 month old APPPS1 and APP-KI mice. Hoechst (HOE, blue) was used for nuclear staining. Yellow arrows indicate intracellular fibrillar Aβ within APPPS1 microglia. Scale bar: 10 μm. (**B**) Quantification of IBA1 positive cells recruited to amyloid plaques in 3 month old APPPS1 and APP-KI mice. Microglial numbers are

*Figure 9 continued on next page*

Figure 9 continued

normalized to the plaque area. (C) Quantification of plaque size in 3 month old APPPS1 and APP-KI mice. (D) Immunohistochemical analysis of IBA1 (green) and CD68 (red) positive microglial cells recruited to Aβ plaques (magenta) in 3 month old APPPS1 and APP-KI mice. Hoechst (HOE, blue) was used for nuclear staining. Scale bar: 20 µm. (E) Quantification of CD68 coverage per single plaque in 3 month old APPPS1 and APP-KI mice. CD68 coverage is normalized to the plaque area. (F) Quantification of CD68 coverage per microglia in 3 month old APPPS1 and APP-KI mice. CD68 coverage is normalized to the number of IBA1 positive cells recruited to amyloid plaque. (G) Immunohistochemical analysis of microglia (IBA1, green) associated to dystrophic neurites (APP, red) in 3 month old APPPS1 and APP-KI mice. Aβ plaque was visualized using an antibody against Aβ (magenta) and Hoechst (HOE, blue) was used for nuclear staining. Scale bar: 10 µm. (H) Quantification of area covered by dystrophic neurites in 3 month old APPPS1 and APP-KI mice. (I) Quantification of IBA1 positive cells associated to dystrophic neurites in 3 month old APPPS1 and APP-KI mice. Number of microglia quantified per single plaque are normalized to its corresponding dystrophic neurite area. Values in (B, C), (H), and (I), are expressed as the mean of N = 5 (APPPS1) and N = 4 (APP-KI) mice ± SD and in (E and F) as the mean of N = 3 mice per genotype ± SD (*p<0.05; **p<0.01, unpaired two-tailed Student's T-test; n.s.: not significant). (J). Mean Fluorescence Intensity (MFI) of *E. coli*-pHrodo signal within the CD11b positive cells in 3 and 6 month old APPPS1, APP-KI and WT mice. CytoD was used to inhibit phagocytosis and serves as a negative control. a.u.: arbitrary units. (K). Percentage of CD11b and *E. coli*-pHrodo double positive cells from the total CD11b positive population. In (J and K), values for the 3 month old group are expressed as the mean of N = 3 mice per genotype ± SD and for the 6 month old group as the mean of N = 2 mice per genotype ± SD (***p<0.001, Two-way ANOVA, Dunnett's multiple comparison test; n.s.: not significant).

The online version of this article includes the following source data and figure supplement(s) for figure 9:

**Source data 1.** Phagocytosis is impaired earlier in APPPS1 compared to APP-KI microglia.

**Figure supplement 1.** pE3-Aβ does not correlate with early microglial changes in 3 month old APPPS1 mice.

relevance for AD pathology in humans (*Boza-Serrano et al., 2019*). Further evidence that some of the presented proteomic alterations of rodent microglia may be relevant for human disease is given by the detection of up-regulated early/middle microglial MARPs, including CD68, TREM2 and ITGAX in microglia surrounding amyloid plaques in postmortem AD brains (*Hopperton et al., 2018*). However, a recent study suggested little overlap between DAM and homeostatic profiles of rodent microglia and human AD patients, implicating a limitation of mouse models of AD (*Zhou et al., 2020*). Of note, microglial signatures have mostly been studied in amyloidosis mouse models and amyloid triggered microglial alterations cannot be directly compared with human microglia from advanced AD stages that are exposed to both amyloid and tau pathology. As microglia emerge as a promising therapeutic target in AD, additional MARP signatures should be validated in human tissue. In particular, early MARPs that are strongly increased in both AD mouse models may serve as a resource to identify novel AD biomarkers and more specific microglial positron emission tomography (PET) tracers that are urgently needed to monitor microglial reactivity in vivo (*Edison et al., 2018*; *Hemonnot et al., 2019*). Middle and late MARPs reveal a decrease of microglial homeostatic functions affecting chemotaxis, cell migration and phagocytosis (e.g., CX3CR1, SYK, P2RY12, BIN2, TFEB and TFE3) and thus mark AD progression.

It is still being discussed which is the main trigger for microglial recruitment to amyloid plaques and their molecular switch from a homeostatic to a neurodegenerative phenotype (*Hemonnot et al., 2019*; *Jung et al., 2015*; *Krasemann et al., 2017*). Our study proposes that microglial recruitment to Aβ deposits and their corresponding disease-associated proteomic alterations may be triggered by fibrillar Aβ. This response could be mediated by dense core fibrillar amyloid and/or smaller fibrillar oligomers that have been proposed as neurotoxic species (*Haass and Selkoe, 2007*). Of note, microglial recruitment did not correlate with neuritic dystrophies as we detected prominent neuritic dystrophies in the APP-KI mice that bear less fibrillar Aβ. Similar plaque morphology with less fibrillar Aβ, as observed in APP-KI mice, has also been reported in AD mice deficient for TREM2 or APOE that also have less microglial cells recruited to amyloid plaques and display prominent neuritic dystrophies (*Parhizkar et al., 2019*; *Sala Frigerio et al., 2019*; *Ulrich et al., 2018*; *Wang et al., 2015*; *Yuan et al., 2016*). APOE may have a dual role and control the transcriptional/translational response of microglia to amyloid as well as amyloid plaque compactness that directs microglial recruitment and thus creates a regulatory feedback-loop. These findings are strengthened by the relevance of *ApoE* and *Trem2* as genetic risk factors of AD (*Karch and Goate, 2015*). Fibrillar Aβ as the trigger for microglial recruitment is also supported by the human pathology where neuritic plaques in AD brains were found surrounded by microglia. In contrast, microglial clustering was not detected at diffuse plaques lacking fibrillar Aβ cores (*D'Andrea et al., 2004*).

Although DAM signatures, that include upregulation of phagocytosis and lysosomal genes, have been suggested as a protective response (*Keren-Shaul et al., 2017*), there is still a lack of direct experimental evidence linking DAM profiles to improved microglial phagocytic function. In addition, it has been proposed that MGnD microglia may represent a dysfunctional phenotype (*Krasemann et al., 2017*). Importantly, our study demonstrates a functional link between proteomic changes and reduced phagocytosis by AD microglia. This is in agreement with Aβ−dependent early phagocytic dysfunction of APPPS1 microglia reported previously (*Krabbe et al., 2013*). Our study shows that APPPS1 microglia start acquiring early MARPs at the age of 3 months, which is already accompanied by reduced phagocytic function. In contrast, less altered proteomic signatures of 3 month old APP-KI microglia correlated with preserved phagocytic function. Pronounced MARP signatures that appeared later in APP-KI microglia (6 months) were subsequently in accordance with phagocytic impairments. Therefore, differences in plaque fibrillization in both mouse models did not only affect microglial recruitment and activation, but also the phagocytic function of microglia. This functional link should be examined further using physiological substrates of microglia such as Aβ, myelin or synaptosomes (*McQuade and Blurton-Jones, 2019*).

Reduced phagocytosis of AD microglia might be related to observed proteomic alterations in lysosomal proteins or cell receptors. TREM2, which we found increased in both mouse models, plays an important role in phagocytosis as mutations of TREM2 related to AD and FTLD impair phagocytic activity of microglia (*Kleinberger et al., 2014*). However, up-regulation of the TREM2/APOE axis involves up-regulation of many phagocytic or lysosomal proteins (e.g., cathepsins or CD68) that are part of MARPs and altered in APPPS1 and APP-KI microglia. Similarly, also transcriptional analysis revealed a downregulation of homeostatic and upregulation of DAM or MGnD program within microglial cells in the vicinity of Aβ plaques (*Keren-Shaul et al., 2017*; *Krasemann et al., 2017*). The increase in lysosomal or phagocytic gene signatures may reflect a compensatory mechanism initiated as a response of microglia to Aβ accumulation in order to enhance phagocytic function. Eventually, this limited microglial response fails to translate into improved Aβ clearance capability.

Phagocytosis might also be altered through differential regulation of toll like receptors (TLR). Among the TLRs, TLR2, an Aβ binding receptor (*Liu et al., 2012*; *McDonald et al., 2016*), showed the strongest increase with age while TLR9 was significantly reduced in APPPS1 and APP-KI mice. Along these lines, TLR2 deficiency reduced the inflammatory response of microglia to Aβ42, but increased Aβ phagocytosis in cultured microglia (*Liu et al., 2012*) while TLR9 is associated with improved Aβ clearance (*Scholtzova et al., 2009*). Thus, differential regulation of TLRs might contribute to the reduced phagocytic activity of aged APPPS1 microglia (*Daria et al., 2017*).

Additionally, many purinergic receptors (e.g., P2RX7, P2RY12 or P2RY13), which are important regulators of chemotaxis, phagocytosis, membrane polarization, and inflammatory signaling and thus emerged as possible microglial targets in AD (*Calovi et al., 2019*; *Hemonnot et al., 2019*), were found down-regulated in both AD mouse models. P2RY12 is regarded as a marker for ramified non-inflammatory microglia (*Mildner et al., 2017*) that is reduced in response to Aβ plaques and therefore represents a homeostatic microglial marker (*Keren-Shaul et al., 2017*; *Krasemann et al., 2017*). In contrast, P2RX4, a purinergic receptor that is likely to be involved in shifting microglia towards a pro-inflammatory phenotype (*Calovi et al., 2019*) or myelin phagocytosis (*Zabala et al., 2018*) had an increased abundance in both AD models. Taken together, our data emphasize alterations of purinergic receptor signaling in AD microglia that may regulate a morphological change towards amoeboid microglia with reduced motility and increased pro-inflammatory activity.

Our study confirms that both mouse models are valuable tools for studying Aβ−induced pathological changes of microglia that are remarkably comparable at advanced stages of amyloidosis. However, the observed differences in the dynamics of early, middle and late MARPs in APPPS1 and APP-KI mice should be considered for the design of pre-clinical studies of microglial repair and will require different time windows for microglial modulation.

In conclusion, we tracked pathological alterations of microglia in two AD mouse models using a proteomic approach. Our work demonstrates that microglial alterations are triggered as a response to Aβ deposition as pre-deposition stages do not reveal proteomic alterations. The conversion to MARPs is supported by changes in TREM2-APOE regulation mechanism. AD microglia display pronounced interferon stimulation, increased antigen presentation, alterations in cell surface receptors, lipid homeostasis and metabolism. These proteomic changes in microglia seem to occur as a response to fibrillar Aβ and are reflected in amoeboid microglial morphology and impaired

phagocytic capacity. Finally, our proteomic dataset serves a valuable research resource providing information on microglial alterations over different stages of Aβ deposition that can be used to monitor therapeutic efficacy of microglial repair strategies.

# Materials and methods

**Key resources table**

| Reagent type (species) or resource | Designation | Source or reference | Identifiers | Additional information |
|---|---|---|---|---|
| Genetic reagent (*Mus musculus*) | APPPS1 | DOI: 10.1038/sj.embor.7400784 | RRID:MGI:5313568 | Heterozygous (C57BL/6 background) |
| Genetic reagent (*Mus musculus*) | APP$^{NL-G-F}$ (APP-KI) | DOI: 10.1038/nn.3697 | RRID:MGI:6160916 | Homozygous (C57BL/6 background) |
| Antibody | Anti-IBA1 (rabbit polyclonal) | Wako | Cat#:019–19741, RRID:AB_839504 | IHC (1:500) |
| Antibody | Anti-IBA1 (goat polyclonal) | Abcam | Cat#:ab5076, RRID:AB_2224402 | IHC (1:500) |
| Antibody | Anti-Amyloid beta NAB228 (mouse monoclonal) | Santa Cruz | Cat#:sc-32277, RRID:AB_626670 | IHC (1:2000) |
| Antibody | Anti-CD68 (rat monoclonal) | Bio-Rad | Cat#:MCA1957G, RRID:AB_324217 | IHC (1:500) WB (1:1000) |
| Antibody | Anti-TREM2 (sheep polyclonal) | R and D Systems | Cat#:AF1729, RRID:AB_354956 | IHC (1:50) |
| Antibody | Anti-Amyloid Y188 (rabbit monoclonal) | Abcam | Cat#:1565–1, RRID:AB_562042 | IHC (1:2000) |
| Antibody | Anti-CLEC7a (rat monoclonal) | Invivogen | Cat#:mabg-mdect, RRID:AB_2753143 | IHC (1:50) |
| Antibody | Anti-TMEM119 (rabbit monoclonal) | Abcam | Cat#:ab209064, RRID:AB_2800343 | IHC (1:200) |
| Antibody | Anti-APOE HJ6.3 (mouse monoclonal) | DOI: 10.1084/jem.20121274 | | IHC (Biotinylated, 1:100) |
| Antibody | Anti-Amyloid beta 3552 (rabbit polyclonal) | DOI: 10.1523/JNEUROSCI.05.2006 | | IHC (1:5000), WB (1:2000) |
| Antibody | Anti-pE3-Amyloid beta J8 (mouse monoclonal) | DOI: 10.3390/molecules23040924 | | IHC (1:500) |
| Antibody | Anti-TREM2 5F4 (rat monoclonal) | DOI: 10.15252/emmm.201606370 | | WB (1:10) |
| Antibody | Anti-APOE (goat polyclonal) | Millipore | Cat#:AB947, RRID:AB_2258475 | WB (1:1000) |
| Antibody | Anti-CSF1R (rabbit polyclonal) | Cell Signaling | Cat#:3152, RRID:AB_2085233 | WB (1:1000) |
| Antibody | Anti-FABP5 (goat polyclonal) | R and DSystems | Cat#:AF1476, RRID:AB_2293656 | WB (1:400) |
| Antibody | Anti-GAPDH (mouse monoclonal) | Abcam | Cat#:ab8245, RRID:AB_2107448 | WB (1:2000) |
| Antibody | Anti-CD11b-APC-Cy7 (rat monoclonal) | BDBiosciences | Cat#:557657, RRID:AB_396772 | FACS (1:200) |

## Animals

Male and female mice of the hemizygous APPPS1 mouse line overexpressing human APP$_{KM670/671NL}$ and PS1$_{L166P}$ under the control of the *Thy1* promoter (*Radde et al., 2006*), homozygous *App$^{NL-G-F}$* mouse line (*Saito et al., 2014*) and the C57BL/6J (WT) line were used in this study. Mice were group housed under specific pathogen-free conditions. Mice had access to water and standard mouse chow (Ssniff Ms-H, Ssniff Spezialdiäten GmbH, Soest, Germany) ad libitum and were kept under a

12/12 hr light-dark cycle in IVC System Typ II L-cages (528 cm$^2$) equipped with solid floors and a layer of bedding. All animal experiments were performed in compliance with the German animal welfare law and have been approved by the government of Upper Bavaria.

## Isolation of primary microglia

Primary microglia were isolated from mouse brains (cerebrum) using MACS technology (Miltenyi Biotec) according to manufacturer's instructions and as previously described (*Daria et al., 2017*). Briefly, olfactory bulb, brain stem and cerebellum were removed and the remaining cerebrum was freed from meninges and dissociated by enzymatic digestion using a Neural Tissue Dissociation Kit P (Miltenyi Biotec). Subsequently, mechanical dissociation was performed by using three fire-polished glass Pasteur pipettes of decreasing diameter. CD11b positive microglia were magnetically labelled using CD11b MicroBeads, loaded onto a MACS LS Column (Miltenyi Biotec) and subjected to magnetic separation, resulting in CD11b-enriched (microglia-enriched) and CD11b-depleted (microglia-depleted) fractions. Obtained microglia-enriched pellets were either washed twice with HBSS (Gibco) supplemented with 7 mM HEPES, frozen in liquid nitrogen and stored at −80℃ for biochemical or mass spectrometry analysis, or resuspended in microglial culturing media and used for phagocytosis assay as described below.

## Sample preparation for mass spectrometry

Microglia were isolated from three transgenic animals and their three corresponding WT controls for each age and genotype. We included both male and female mice into proteomic analysis and their distribution is outlined in *Supplementary file 3*. Microglia-enriched pellets from individual animals were analysed separately and lysed in 200 µL of STET lysis buffer (50 mM Tris, 150 mM NaCl, 2 mM EDTA, 1% Triton, pH 7.5) at 4℃ with intermediate vortexing. The samples were centrifuged for 5 min at 16000 x *g* at 4℃ to remove cell debris and undissolved material. The supernatant was transferred to a LoBind tube (Eppendorf) and the protein concentration estimated using the Pierce 660 nm protein assay (ThermoFisher Scientific). A protein amount of 15 µg was subjected to tryptic protein digestion applying the the filter aided sample preparation protocol (FASP) (*Wiśniewski et al., 2009*) using Vivacon spin filters with a 30 kDa cut-off (Sartorius). Briefly, proteins were reduced with 20 mM dithiothreitol and free cystein residues were alkylated with 50 mM iodoacetamide (Sigma Aldrich). After the urea washing steps, proteins were digested with 0.3 µg LysC (Promega) for 16 hr at 37℃ followed by a second digestion step with 0.15 µg trypsin (Promega) for 4 hr at 37℃. The peptides were eluted into collection tubes and acidified with formic acid (Sigma Aldrich). Afterwards, proteolytic peptides were desalted by stop and go extraction (STAGE) with self-packed C18 tips (Empore C18 SPE, 3M) (*Rappsilber et al., 2003*). After vacuum centrifugation, peptides were dissolved in 20 µL 0.1% formic acid (Biosolve) and indexed retention time peptides were added (iRT Kit, Biognosys).

## Liquid chromatography – tandem mass spectrometry analysis

For label-free quantification (LFQ) of proteins, peptides were analyzed on an Easy nLC 1000 or 1200 nanoHPLC (Thermo Scientific) which was coupled online via a Nanospray Flex Ion Source (Thermo Sientific) equipped with a PRSO-V1 column oven (Sonation) to a Q-Exactive HF mass spectrometer (Thermo Scientific). An amount of 1.3 µg of peptides was separated on in-house packed C18 columns (30 cm x 75 µm ID, ReproSil-Pur 120 C18-AQ, 1.9 µm, Dr. Maisch GmbH) using a binary gradient of water (A) and acetonitrile (B) supplemented with 0.1% formic acid (0 min., 2% B; 3:30 min., 5% B; 137:30 min., 25% B; 168:30 min., 35% B; 182:30 min., 60% B) at 50℃ column temperature.

For DDA, full MS scans were acquired at a resolution of 120000 (m/z range: 300–1400; automatic gain control (AGC) target: 3E+6). The 15 most intense peptide ions per full MS scan were selected for peptide fragmentation (resolution: 15000; isolation width: 1.6 m/z; AGC target: 1E+5; normalized collision energy (NCE): 26%). A dynamic exclusion of 120 s was used for peptide fragmentation.

For DIA, one scan cycle included a full MS scan (m/z range: 300–1400; resolution: 120000; AGC target: 5E+6 ions) and 25 MS/MS scans covering a range of 300–1400 m/z with consecutive m/z windows (resolution: 30000; AGC target: 3E+six ions; *Supplementary file 4*). The maximum ion trapping time was set to 'auto'. A stepped normalized collision energy of 26 ± 2.6% was used for fragmentation.

Microglia from APPPS1 mice were analyzed using DDA and DIA for method establishement. Microglia from APPPS1 and APP-KI mice were compared using DIA as it outperformed DDA.

## Mass spectrometric LFQ and data analysis

For data acquired with DDA, the data was analyzed with the software Maxquant (maxquant.org, Max-Planck Institute Munich) version 1.6.1.0 (*Cox et al., 2014*). The MS data was searched against a reviewed canonical fasta database of *Mus musculus* from UniProt (download: November the 1st 2017, 16843 entries) supplemented with the sequence of human APP with the Swedish mutant and the iRT peptides. Trypsin was defined as a protease. Two missed cleavages were allowed for the database search. The option first search was used to recalibrate the peptide masses within a window of 20 ppm. For the main search peptide and peptide fragment mass tolerances were set to 4.5 and 20 ppm, respectively. Carbamidomethylation of cysteine was defined as static modification. Acetylation of the protein N-term as well as oxidation of methionine was set as variable modification. The FDR for both peptides and proteins was set to 1%. The 'match between runs' option was enabled with a matching window of 1.5 min. LFQ of proteins required at least one ratio count of unique peptides. Only unique peptides were used for quantification. Normalization of LFQ intensities was performed separately for the age groups because LC-MS/MS data was acquired in different batches.

A spectral library was generated in Spectronaut (version 12.0.20491.11, Biognosys) (*Bruderer et al., 2015*) using the search results of Maxquant of the APPPS1 dataset. The library includes 122542 precursor ions from 91349 peptides, which represent 6223 protein groups. The DIA datasets of both mouse models were analyzed with this spectral library (version 12.0.20491.14.21367) with standard settings. Briefly, the FDR of protein and peptide identifications was set to 1%. LFQ of proteins was performed on peptide fragment ions and required at least one quantified peptide per protein. Protein quantification was performed on maximum three peptides per protein group. The data of APPPS1 microglia was organized in age-dependent fractions to enable separate normalization of the data. All LC-MS/MS runs of the APP-KI dataset were normalized against each other because all samples were analyzed in randomized order in one batch.

The protein LFQ reports of Maxquant and Spectronaut were further processed in Perseus (*Tyanova et al., 2016*). The protein LFQ intensities were log2 transformed and log2 fold changes were calculated between transgenic and WT samples separately for the different age groups and mouse models. Only proteins with a consistent quantification in all samples of an age group were considered for statistical testing. A two-sided Student's t-test was applied to evaluate the significance of proteins with changed abundance (from log2 fold change of transgenic *versus* WT microglia per age group). Additionally, a permutation based FDR estimation (threshold: FDR = 5%, $s_{0x00A0} = 0.1$) was used to perform multiple hypothesis correction (*Tusher et al., 2001*). A log2 fold change larger than 0.5, or smaller than −0.5, a p-value less than 0.05 and significant regulation after FDR filtering were defined as regulation thresholds criteria. The same thresholds were used for the comparison with transcriptomics data.

Gene ontology enrichment analysis was performed with the web-tool DAVID (version 6.8) (*Huang et al., 2009a*; *Huang et al., 2009b*) using GO_FAT terms. Up- and down-regulated early, middle and advanced MARPs were clustered separately for biological process, cellular component, and molecular function with all 5500 proteins, consistently quantified in APPPS1 and APP-KI microglia, as a customized background. A medium classification stringency was applied. An enrichment score of 1.3 was defined as threshold for cluster enrichment.

## Biochemical characterization of brain tissue and isolated microglia

RIPA lysates were prepared from brain hemispheres, centrifuged at 100000 x *g* (60 min at 4°C) and the remaining pellet was homogenized in 0.5 mL 70% formic acid. The formic acid fraction was neutralized with 20 × 1 M Tris-HCl buffer at pH 9.5 and used for Aβ analysis. For Aβ detection, proteins were separated on Tris-Tricine gels (10–20%, Thermo Fisher Scientific), transferred to nitrocellulose membranes (0.1 μm, GE Healthcare) which were boiled for 5 min in PBS and subsequently incubated with the blocking solution containing 0.2% I-Block (Thermo Fisher Scientific) and 0.1% Tween 20 (Merck) in PBS for 1 hr, followed by overnight incubation with rabbit polyclonal 3552 antibody (1:2000, *Yamasaki et al., 2006*). Antibody detection was performed using the corresponding anti-

HRP conjugated secondary antibody (Santa Cruz) and chemiluminescence detection reagent ECL (Thermo Fisher Scientific).

Microglia-enriched pellets were resuspended in 100 µL of STET lysis buffer (composition as described above for mass spectrometry, supplemented with protease and phosphatase inhibitors), kept on ice for 20 min and then sonicated for 4 cycles of 30 s. Cell lysates were then centrifuged at 9600 x $g$ (5 min. at 4°C) and pellets discarded. Protein concentration was quantified using Bradford assay (Biorad) according to manufacturer instructions. 10 µg of two independent microglial lysates per genotype were loaded on a bis-tris acrylamide gel (8% or 12%) and subsequently blotted onto either a PVDF or nitrocellulose membrane (Millipore) using the following antibodies: TREM2 (1:10, clone 5F4, *Xiang et al., 2016*); APOE (1:1000, AB947 Millipore); CD68 (1:1000, MCA1957GA, AbD-serotec); CSF1R (1:1000, 3152, Cell Signaling) and FABP5 (1:400, AF1476, R and DSystems). Blots were developed using horseradish peroxidase-conjugated secondary antibodies (Promega) and the ECL chemiluminescence system (Amersham) or SuperSignal West Pico PLUS (Thermo Scientific). An antibody against GAPDH (1:2000, ab8245, Abcam) was used as loading control.

## Immunohistochemistry

We analyzed 3 and 12 month old APPPS1 and APP-KI mice of both sexes, as outlined in *Supplementary file 5*. Mice were anesthetized i.p. with a mixture of ketamine (400 mg/kg) and xylazine (27 mg/kg) and transcardially perfused with cold 0.1M PBS for 5 min followed by 4% paraformaldehyde (PFA) in 0.1 M PBS for 15 min. Brains were isolated and postfixed for 20 min in 4% PFA in 0.1 M PBS and transferred to 30% sucrose in 0.1 M PBS for cryopreservation. Brains were embedded in optimal cutting temperature compound (Tissue-Tek O.C.T., Sakura), frozen on dry ice and kept at −80°C until sectioning. 30 µm coronal brain sections were cut using a cryostat (CryoSTAR NX70, Thermo Scientific) and placed in 0.1 M PBS until staining. Alternatively, sections were kept in anti-freezing solution (30% glycerol, 30% ethylenglycol, 10% 0.25 M $PO_4$ buffer, pH 7.2–7.4% and 30% $dH_2O$) at −20°C and briefly washed in 0.1M PBS before staining. Briefly, free-floating sections were permeabilized with 0.5% Triton-PBS (PBS-T) for 30 min, blocked either in 5% normal goat Serum or 5% donkey serum in PBS-T for 1 hr and incubated overnight at 4°C in blocking solution with the following primary antibodies: IBA1 (1:500, 019–19741,Wako), IBA1 (1:500, ab5076, Abcam), NAB228 (1:2000, sc-32277, Santa Cruz), CD68 (1:500, MCA1957GA, Bio-Rad), TREM2 (1:50, AF1729, R and DSystems), APP-Y188 (1:2000, ab32136, Abcam), CLEC7a (1:50, mabg-mdect, Invivogen), TMEM119 (1:200, ab209064, Abcam), APOE-biotinylated (HJ6.3, 1:100, [*Kim et al., 2012*]), 3552 (1:5000, [*Yamasaki et al., 2006*]) and pE3-Aβ (J8, 1:500, [*Hartlage-Rübsamen et al., 2018*]). pE3-Aβ immunostaining required heat antigen retrieval (25 min at 95°C) in citrate buffer (10 mM, pH 6.0) prior blocking. After primary antibody incubation, brain sections were washed 3 times with PBS-T and incubated with appropriate fluorophore-conjugated or streptavidine-fluorophore conjugated (for APOE biotinylated antibody) secondary antibodies (1:500, Life Technologies) together with nuclear stain Hoechst 33342 (1:2000, H3570, Thermo Fisher Scientific), for 2 hr at room temperature (RT). Fibrillar dense core plaques were stained with ThR (Sigma Aldrich, 2 µM solution in PBS) for 20 min in the dark at RT (after secondary antibody staining). Sections were subsequently washed 3 times with PBS-T, mounted onto glass slides (Thermo Scientific), dried in the dark for at least 30 min, mounted using Gel Aqua Mount media (Sigma Aldrich) and analyzed by confocal microscopy.

## Image acquisition, analysis and quantifications

3 month old APPPS1 and APP-KI mice were analyzed for dystrophic neurites, plaque size, microglial recruitment and CD68 and total Aβ coverage. pE3-Aβ coverage was analyzed in 12 month old APPPS1 and APP-KI mice. All quantification analysis were performed in a blinded manner and included at least three mice per genotype. Mice sex and numbers of biological and technical replicates for all immunohistological experiments are summarized in *Supplementary file 5*.

For the analysis of microglial recruitment, plaque and dystrophic neurite size and CD68 coverage, 30 z-stack images (10–12 µm thick) of randomly selected plaques from neocortical regions were acquired per experiment using a confocal microscope (63X water objective with 2x digital zoom, 600 Hz, Leica TCS SP5 II). Microscopy acquisition settings were kept constant within the same experiment. Maximal intensity projection pictures from every z-stack were created using ImageJ software and for every image, a defined region of interest (ROI) was manually drawn around every plaque

(including microglia recruited -in contact- to the plaque). APP (Y188 antibody) and CD68 coverage area were quantified using the 'Threshold' and 'Analyze Particles' (inclusion size of 1-Infinity) functions from ImageJ software (NIH) within the ROI. The area covered by CD68 was normalized to the total Aβ plaque area (NAB228 antibody) or was divided by the number of microglia (IBA1 positive cells) recruited to the plaque within the ROI. The absolute values of area covered by neuritic dystrophies or Aβ plaques are represented in square micrometers ($\mu m^2$). Microglial recruitment to plaques was quantified by counting the number of microglia (IBA1 positive cells) around amyloid plaques through the z-stack images within the defined ROI using the cell counter function of ImageJ software. Number of microglial cells at amyloid plaques was normalized to the area covered by Aβ (NAB228 antibody) or by dystrophic neurites (APP, Y188 antibody) and expressed as number of microglial cells per $\mu m^2$ of Aβ plaque.

For the quantification of total Aβ and pE3-Aβ coverage at 3 and 12 months of age, respectively, 18 images were systematically taken from comparable neocortical regions using a confocal microscope (20X dry objective, 600 Hz, Leica TCS SP5 II). Quantification of Aβ and pE3-Aβ areas was performed using a self-programmed macro with ImageJ software outlined in *Supplementary file 6A and B*, respectively.

Representative images from microglial recruitment analysis (IBA1 positive cells and CD68 coverage) were taken using the confocal microscope (63X water objective with 2x digital zoom, 400 Hz, Leica TCS SP5 II). Representative images of microglia polarized towards amyloid cores and of microglial recruitment and dystrophic neurite size were taken using a 63X confocal water objective with 3x digital zoom.

For immunohistological validation of the proteome and analysis of amyloid pathology, representative images from comparable neocortical regions were taken by confocal microscopy using the same settings for all three different genotypes (WT, APPPS1 and APP-KI). Low magnification pictures were taken with 20X dry confocal objective with 2x digital zoom and higher magnification ones with 63X confocal water objective with 3x digital zoom (400 Hz, Leica TCS SP5 II). Images of Aβ pathology (NAB228 antibody) were taken with a tile scan system covering comparable cortico-hippocampal regions (10X confocal dry objective, 400 Hz, Leica TCS SP5 II). Representative images of Aβ composition and microglia (NAB228, ThR and IBA1) and pE3-Aβ were taken with a confocal 20X dry objective (400 Hz, Leica TCS SP5 II).

## LCO staining and spectral analysis

3 and 12 month old APPPS1 and APP-KI mice were analysed using LCOs. Information including mice sex and biological and technical replicates is outlined in *Supplementary file 5*. LCO analysis was performed as previously described (*Rasmussen et al., 2017*). Briefly free-floating brain sections from PFA-perfused mice (as described in *Immunohistochemistry*) were incubated with two LCOs, qFTAA and hFTAA (2.4 μM qFTAA and 0.77 μM hFTAA in PBS), for 30 min at RT in the dark and subsequently washed 3 times with PBS-T. Sections were mounted, air-dried and mounted with Dako mounting medium. Randomly chosen images from neocortex (4–10 images per mouse) were acquired with a confocal 40X oil objective on a Zeiss LSM 510 META confocal microscope equipped with an argon 458 nm laser for excitation and a spectral detector. Emission spectra were acquired from 470 to 695 nm and values of each plaque were normalized to their respective maxima. For each mouse, 20 plaques were randomly chosen from the images, normalized values were averaged for each mouse and the mean was taken for each mouse line (spectrum). The ratio of the intensity of emitted light at the blue-shifted peak (502 nm from qFTAA) and red-shifted peak (588 nm from hFTAA) was used as a parameter for spectral distinction of different Aβ conformations in plaque cores.

## Microglial phagocytosis of *E. coli* particles

Microglial phagocytosis was performed similarly as previously described (*Kleinberger et al., 2014*). Microglia isolated from 3 or 6 month old APPPS1, APP-KI and WT mice were plated onto 24 well plate at a density of $2 \times 10^5$ cells per well and cultured for 24 hr in a humidified 5% $CO_2$ incubator at 36.5°C in DMEM/F12 media (Invitrogen) supplemented with 10% heat inactivated FCS (Sigma), 1% Penicillin-Streptomycin (Invitrogen) and 10 ng/mL GM-CSF (R&DSystems). After 24 hr, plating media were replaced with fresh media. After 5 days in culture, microglia were incubated with 50 μL

of *E. coli* particle suspension (pHrodo Green *E. coli* BioParticles, P35366, Invitrogen) for 60 min. Cytochalasin D (CytoD, 10 μM, from 10 mM stock in DMSO) was used as phagocytosis inhibitor and added 30 min prior to addition of bacterial particles. Bacteria excess was washed four times with PBS (on ice) and microglia attached to the plate were incubated with CD11b-APC-Cy7 antibody (1:200, clone M1/70, 557657, BD) in FACS buffer (PBS supplemented with 2 mM EDTA and 1% FBS) for 30 min at 4°C. Microglia were then washed twice with PBS, scraped off in FACS buffer and analyzed by flow cytometry. Information including mice sex and biological and technical replicates is outlined in *Supplementary file 7*.

## FACS analysis

For the microglial isolation quality control, around 12000 cells from a CD11b-enriched and CD11b-depleted fractions were stained in suspension with CD11b-APC-Cy7 antibody (1:200, clone M1/70, 557657, BD) in FACS buffer for 30 min at 4°C. After several washes with PBS, microglia were resuspended in FACS buffer for analysis. Propidium Iodide (PI) staining was done 10 min prior to FACS analysis. Flow cytometric data was acquired on a BD FACSverse flow cytometer by gating according to single stained and unstained samples and analyzed using FlowJo software (Treestar). Mean fluorescent intensity (MFI) is represented as the geometric mean of the according fluorochrome.

## Immunohistochemical and FACS data analysis

Immunohistochemical and FACS results are presented as mean ± standard deviation (± SD) from at least three independent experiments with the exception of the phagocytic assay at 6 months where two independent experiments were performed (*Figure 9J and K*). Statistical significance (p-value) was calculated using the unpaired two-tailed Student's *t*-test. Phagocytic assay was analyzed by the Dunnett's multiple comparison test of the Two-way ANOVA. Both statistical analysis were performed in GraphPad Prism. P-value of <0.05 was considered statistically significant (*; p<0.05, **; p<0.01, ***; p<0.001 and ****; p<0.0001, n.s.=not significant).

## Data availability

The mass spectrometry proteomics data have been deposited to the ProteomeXchange Consortium via the PRIDE partner repository (*Perez-Riverol et al., 2019*) with the dataset identifier PXD016075.

## Acknowledgements

We thank Allison Morningstar and Lina Dinkel for critically reading the manuscript. The authors are grateful to David Holtzman (Washington University School of Medicine, St Louis, Missouri, USA) for sharing the ApoE antibody, Stephan Schilling (Fraunhofer IZI-MWT, Halle/Saale, Germany) for the gift of the mouse monoclonal anti-pE-A$\beta$ antibody J8, K Peter Nilsson (Linköping University, Sweden) for providing the LCOs and Mathias Jucker (Hertie-Institute for Clinical Brain Research, University of Tübingen, Germany) for providing the APPPS1 mice and assisting with the LCOs analysis. We thank Haike Hampel for excellent technical assistance.

## Additional information

### Funding

| Funder | Grant reference number | Author |
| --- | --- | --- |
| Alzheimer Forschung Initiative | 18014 | Sabina Tahirovic |
| Deutsche Forschungsgemeinschaft | EXC 2145 SyNergy ID 390857198 | Arthur Liesz<br>Jochen Herms<br>Christian Haass<br>Stefan F Lichtenthaler |
| Deutsche Forschungsgemeinschaft | Koselleck Project HA1737/16-1 | Christian Haass |
| H2020 European Research Council | ERC-StG 802305 | Arthur Liesz |

| Helmholtz-Gemeinschaft | ZT-0027 | Christian Haass |
| Bundesministerium für Bildung und Forschung | JPND PMG-AD | Christian Haass Stefan F Lichtenthaler |
| Bundesministerium für Bildung und Forschung | CLINSPECT-M | Stefan F Lichtenthaler |
| NCL Foundation | 01133 | Sabina Tahirovic |

The funders had no role in study design, data collection and interpretation, or the decision to submit the work for publication.

### Author contributions
Laura Sebastian Monasor, Formal analysis, Validation, Investigation, Visualization, Methodology, Writing - original draft; Stephan A Müller, Resources, Software, Formal analysis, Investigation, Visualization, Methodology, Writing - original draft; Alessio Vittorio Colombo, Stefan Roth, Formal analysis, Investigation, Methodology, Writing - review and editing; Gaye Tanrioever, Formal analysis, Investigation, Methodology; Jasmin König, Anna Berghofer, Investigation, Methodology, Writing - review and editing; Arthur Liesz, Supervision, Methodology, Writing - review and editing; Anke Piechotta, Resources, Writing - review and editing; Matthias Prestel, Formal analysis, Writing - review and editing; Takashi Saito, Takaomi C Saido, Jochen Herms, Methodology, Writing - review and editing; Michael Willem, Formal analysis, Investigation, Writing - review and editing; Christian Haass, Supervision, Writing - review and editing; Stefan F Lichtenthaler, Sabina Tahirovic, Conceptualization, Supervision, Funding acquisition, Writing - original draft

### Author ORCIDs
Laura Sebastian Monasor (iD) https://orcid.org/0000-0001-7864-7400
Stephan A Müller (iD) https://orcid.org/0000-0003-3414-307X
Stefan F Lichtenthaler (iD) https://orcid.org/0000-0003-2211-2575
Sabina Tahirovic (iD) https://orcid.org/0000-0003-4403-9559

### Ethics
Animal experimentation: This study was performed in compliance with the German animal welfare law. All experimental procedures have been approved by the government of Upper Bavaria and covered by the general institutional allowance for breeding of transgenic animals (#04-26a) and thus did not need separate ethical approval.

### Decision letter and Author response
Decision letter https://doi.org/10.7554/eLife.54083.sa1
Author response https://doi.org/10.7554/eLife.54083.sa2

## Additional files

### Supplementary files
• Supplementary file 1. Comparison of proteomic results of APPPS1 microglia using DDA and DIA including the average peptide IDs, protein IDs, protein quantifications calculated for all samples, numbers of relatively quantified proteins from 1, 3, 6, and 12 months as well as their averages.

• Supplementary file 2. Quantitative proteomic data analysis of APPPS1 (**A**) and APP-KI (**B**) *versus* WT microglia at 1, 3, 6, and 12 months using DIA. The table shows the number of consistently quantified proteins as well as proteins with a significant up- or down-regulation with and without FDR correction. A log2 fold change >0.5 or<−0.5 and a p-value of less than 0.05 were applied as regulation thresholds. The amount of up-and down-regulated proteins with FDR correction is shown as percentage from the total number of quantified proteins.

• Supplementary file 3. Mice sex and biological replicates for the proteomic analysis of microglia.

- Supplementary file 4. Optimized mass to charge (m/z) window distribution for Sequential Window Acquisition of all theoretical Mass Spectra (SWATH-MS) based on DIA.

- Supplementary file 5. Mice sex and biological and technical replicates analyzed by immunohistochemistry.

- Supplementary file 6. Self-programmed macros from ImageJ software used for quantification of the total Aβ coverage (**A**) and pE3-Aβ coverage (**B**). Explanation of functions is delineated in green.

- Supplementary file 7. Mice sex and biological and technical replicates analyzed by FACS.

- Transparent reporting form

### Data availability

The mass spectrometry proteomics data have been deposited to the ProteomeXchange Consortium via the PRIDE partner repository (Perez-Riverol et al., 2019) with the dataset identifier PXD016075.

The following dataset was generated:

| Author(s) | Year | Dataset title | Dataset URL | Database and Identifier |
|-----------|------|---------------|-------------|------------------------|
| Müller SA | 2020 | Microglial proteomic signatures in APPPS1 and APP-KI mice | https://www.ebi.ac.uk/pride/archive/projects/PXD016075 | PRIDE, PXD016075 |

The following previously published dataset was used:

| Author(s) | Year | Dataset title | Dataset URL | Database and Identifier |
|-----------|------|---------------|-------------|------------------------|
| Amit I, Keren-Shaul H, Spinrad A, Weiner A, Matcovitch-Natan O | 2017 | Single cell RNA-seq identifies a unique microglia type associated with Alzheimer's disease [RNA] | https://www.ncbi.nlm.nih.gov/geo/query/acc.cgi?acc=GSE98969 | NCBI Gene Expression Omnibus, GSE98969 |

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
