## [Decision Letter]

**Acceptance summary:**

Monasor and colleagues perform a detailed and systematic analysis of age-related changes in the microglial proteome across two independent mouse models of Alzheimer's disease and show that changes in the microglial proteome are more closely associated with increasing fibrillar amyloid and relatively independent of dystrophic neurite pathology. Their findings suggest plaque-associated microglia exhibit impaired phagocytosis that begins relatively early in the disease process. Interestingly, they find that proteomic changes correlate only partially with prior RNA-seq analyses of AD mice, revealing many additional changes that were not previously detected at the mRNA level, further demonstrating the importance of examining the microglial proteome. Overall this work provides new insight into the microglial responses to plaque pathology and will be a useful resource for the field.

**Decision letter after peer review:**

Thank you for submitting your article "Fibrillar Aβ triggers microglial proteome alterations and dysfunction in Alzheimer mouse models" for consideration by *eLife*. Your article has been reviewed by three peer reviewers, including Beth Stevens as the Reviewing Editor and Reviewer #1, and the evaluation has been overseen by Huda Zoghbi as the Senior Editor. The following individuals involved in review of your submission have agreed to reveal their identity: Mathew Blurton-Jones (Reviewer #2); Tara L Spires-Jones (Reviewer #3).

The reviewers have discussed the reviews with one another, and the Reviewing Editor has drafted this decision to help you prepare a revised submission.

Sebastian Monasor and colleagues report a study examining proteomic changes in microglia in 2 mouse models of Alzheimer's disease, APPPS1 and APP-NL-G-F mice. This work builds upon a solid number of single cell transcriptomic papers showing disease associated changes in microglia in plaque bearing models and human AD brain. Here, the multiple time points and use of proteomics expands what we know about microglial changes in AD models. Authors isolated microglia from mouse brains, optimized a proteomic workflow, then used data independent acquisition for label free quantification of proteins. They observed protein signatures similar to wild-type mice at 1 month of age in both lines. In APPPS1 mice, early microglial protein changes in were observed at 3 months of age with further changes at 6 and 12 months. APP-NL-G-F mice exhibited changes later, starting at 6 months of age.

All reviewers agree that this paper has the potential to be a useful resource given our limited knowledge about the microglia proteome in aging and disease models. There is a nice use of two models that show consistent findings. These datasets contain further confirmation that some of the AD risk genes are acting through changes in protein levels in microglia.

The proteomics datasets could be a valuable resource for the field; however, there are significant concerns about robustness of data and the lack of transparency and details on methods, statistics and analyses that must be addressed as detailed by reviewer 3.

Essential revisions:

1) Provide more information and clarity on methods, experimental design, sample size.

In addition, there were shared concerns raised about rigor and lack of details on statistical analyses and specific tests used on all figures. Please see specific concerns detailed by reviewer 3, each one needs to be addressed.

2) Address possible sex-specific effects and report the number of makes and female mice used in each experiment. Were groups balanced to include equivalent numbers of male and female mice? Were the experiments powered to include a sex-dependent analysis?

3) Is there indeed an inverse correlation between the number of plaque-associated microglia and dystrophic neurite area? Please include a regression analysis of these data across the two mouse lines.

4) Please move these two Venn diagrams from Supplemental figures 2 and 4C into the main figures.

5) A more in-depth characterization of the different amyloid types between the Transgenic and knock in models. The authors highlight differences in fibrillar amyloid and plaque compaction between the models and that proteomic changes more closely correlated with the fibrillar Aβ than the neuritic dystrophy However, would be helpful to see more quantification and examination of oligomeric or N-terminal (pyroglu-Aβ) modified species and to address whether the changes also followed differences in those species.

Reviewer #3:

Strengths:

Overall this is a clearly written paper on an important topic. There is a very nice use of 2 models, and I appreciate that the authors examined changes that were consistent between the 2 lines. These datasets contain nice further confirmation that some of the AD risk genes are acting through changes in protein levels in microglia. The shared proteomics datasets will also be very useful for the field.

Critiques:

I have a substantive concern with the robustness of the data. This may be completely alleviated by more methods details, or there may be some experimental design and analysis problems that need addressing. More details are needed to understand better. Namely:

How many mice were used per group in the proteomic study? Were all microglia from all of the mice in a single group pooled or were they analysed separately then an average taken per group?

I'm concerned by the use of Student’s t-tests in proteomic abundance comparisons as there were multiple testing and multiple genotypes. Is this robust?

What were the sexes of the mice, were they sex balanced, and were there sex effects? It is known that there are sex differences in some microglial phenotypes.

Similarly, in the histology studies, 3 mice per group is not very robust and it looks like the data analysis may be pseudoreplicated. In the Materials and methods, it states that 3 mice were used per group with 6 sections from each mouse and 5 plaques for each section, so 30 images per mouse. Were the images all from neocortex? Were the 6 sections per mouse systematically chosen through the brain and from the same locations in each mouse?

Importantly, in the analysis of Figure 8, it is not clear whether the experimental unit is the individual mouse or individual plaque, e.g. is the n 3 or 90? If the latter, the data are pseudoreplicated and not robust.

What were the sex of the mice in IHC studies and were there sex effects?

---

## [Author Response]

Essential revisions:1) Provide more information and clarity on methods, experimental design, sample size.In addition, there were shared concerns raised about rigor and lack of details on statistical analyses and specific tests used on all figures. Please see specific concerns detailed by reviewer 3, each one needs to be addressed.Concerns detailed by reviewer #3:1.1) How many mice were used per group in the proteomic study? Were all microglia from all of the mice in a single group pooled or were they analysed separately then an average taken per group?

For our proteomic analysis, we used 3 animals per age group and genotype. The extracted microglial lysates were measured and analysed for each animal individually (“Microglia were isolated from 3 transgenic animals and their 3 corresponding WT controls for each age and genotype…Microglia-enriched pellets from individual animals were analysed separately…”). Statistical testing was based on label-free quantification (LFQ) intensities of the 3 individual animals per group and outlined as follows: “The protein LFQ intensities were log2 transformed and log2 fold changes were calculated between transgenic and WT samples separately for the different age groups and mouse models”.

1.2) I'm concerned by the use of Student’s t-tests in proteomic abundance comparisons as there were multiple testing and multiple genotypes. Is this robust?

As mass spectrometry analysis of the two transgenic lines was performed in separate batches, LFQ intensities can only be compared between the transgenic and WT mice of the same age. Since we did not perform multiple comparisons, the t-test is the appropriate choice for statistical analysis. Taking into consideration that our sample size is small (N=3), we applied the rather conservative FDR correction (5%) that reduces false positive findings and controls for the possible inflation of the p-value. Furthermore, we have defined minimal threshold of log2 fold change of 0.5 (≙ 1.41-fold on linear scale) to avoid minor, but statistically significant changes.

1.3) What were the sexes of the mice, were they sex balanced, and were there sex effects? It is known that there are sex differences in some microglial phenotypes.Similarly, in the histology studies, 3 mice per group is not very robust and it looks like the data analysis may be pseudoreplicated. In the Materials and methods, it states that 3 mice were used per group with 6 sections from each mouse and 5 plaques for each section, so 30 images per mouse. Were the images all from neocortex? Were the 6 sections per mouse systematically chosen through the brain and from the same locations in each mouse?

We have included both male and female mice into our analysis. Although we acknowledge sex-specific effects on microglial phenotypes, our experiments are unfortunately not powered to enable sex-dependent analysis. To counteract possible sex-dependent effects, our experimental groups for the proteomic analysis were sex-balanced (with the exception of APPPS1 at 12 months). We have outlined the sex of mice analyzed in all experiments in Supplementary files 3 (proteomics), 5 (IHC) and 7 (FACS) and discussed this limitation of our study as follows: “Although, we included both male and female mice for the analysis of microglial proteome, our study was not powered to detect sex-specific differences that have been reported in microglia (Sala Frigerio et al., 2019)”. For our histological studies, we increased the number of analysed animals (from N=3 to N=4 for APP-KI and N=5 for APPPS1) in our revised manuscript for experiments delineated in Figures 9B, C, H and I. The 6 sections that were analysed per mouse in our IHC studies of amyloid plaques were systematically chosen coronal sections that represent comparable brain regions in each mouse. 30 randomly selected plaques from the neocortex were then analyzed per mouse (subsection “Image acquisition, analysis and quantifications” and Supplementary file 5).

1.4) Importantly, in the analysis of Figure 8, it is not clear whether the experimental unit is the individual mouse or individual plaque, eg is the n 3 or 90? If the latter, the data are pseudoreplicated and not robust.

The experimental unit used for statistical testing in Figure 9 (former Figure 8) is the individual mouse (N=3 or higher in the revised manuscript) and not the individual plaque. This information is outlined in Supplementary file 5 of our revised manuscript.

1.5) What were the sex of the mice in IHC studies and were there sex effects?

As stated above, the sex of all mice analyzed by IHC are outlined in Supplementary file 5. Unfortunately, our experiments did not allow a sex-dependent analysis (Discussion paragraph two).

2) Address possible sex-specific effects and report the number of makes and female mice used in each experiment. Were groups balanced to include equivalent numbers of male and female mice? Were the experiments powered to include a sex-dependent analysis?

As outlined above, our experiments are unfortunately not powered to include a sex-dependent analysis. We acknowledge sex-specific effects on microglia and thus our experimental groups for the proteomic analysis (with the exception of APPPS1 at 12 months) were sex-balanced. We reported numbers of male and female mice analysed in all experiments in Supplementary files 3 (proteomics), 5 (IHC) and 7 (FACS) and discussed this limitation of our study as follows: “Although, we included both male and female mice for the analysis of microglial proteome, our study was not powered to detect sex-specific differences that have been reported in microglia (Sala Frigerio et al., 2019)”.

3) Is there indeed an inverse correlation between the number of plaque-associated microglia and dystrophic neurite area? Please include a regression analysis of these data across the two mouse lines.

As pointed out by the reviewer, we found an increased dystrophic neurite area in the APP-KI compared to the APPPS1 mice and reduced number of microglial cells recruited to this area. This additional quantification analysis is now depicted in new Figure 9I. As suggested by the reviewer, we also performed a regression analysis between the number of plaque-associated microglia and dystrophic neurite area, but could not find an inverse correlation (Author response image 1). Therefore, our data do not provide evidence that more microglia around amyloid plaques are directly responsible for less neuritic damage.

**Author response image 1. sa2fig1:** Low correlation between dystrophic neurite size and microglia recruitment. Correlation and linear regression analysis of dystrophic neurite size and microglia recruitment per plaque in APPPS1 (A) and APP-KI (B) mice. r_s_ indicates the Spearman’s rank correlation coefficient, p<0.05: significant, p≥ 0.05: non significant. Different colors represent biological replicates (APPPS1: N=5 mice, APP-KI: N=4 mice) and the same color is used to show technical replicates.

4) Please move these two Venn diagrams from supplemental figures 2 and 4C into the main figures.

We appreciate this comment and followed the reviewer’s recommendation. A new Figure 4 is now depicting a comparison between our two models (12 months) and Keren-Shaul dataset, in both numbers of identified proteins (Figure 4A) and numbers of unidirectionally regulated proteins (Figure 4B) as described in paragraph seven of subsection “Identification of MARPs as signatures of early, middle and advanced amyloid stages”. Of note, as former supplemental Figure 2D (an overlap of relatively quantified proteins in APPPS1 and APP-KI at 12 months) supports that our proteomic workflow in both models is highly reproducible, we still left this technical information in Figure 1—figure supplement 1D.

5) A more in-depth characterization of the different amyloid types between the Transgenic and knock in models. The authors highlight differences in fibrillar amyloid and plaque compaction between the models and that proteomic changes more closely correlated with the fibrillar Aβ than the neuritic dystrophy However, would be helpful to see more quantification and examination of oligomeric or N-terminal (pyroglu-Aβ) modified species and to address whether the changes also followed differences in those species.

In the revised manuscript we quantified the coverage of total Aβ at 3 months of age in both mouse models (Figure 8C), and found an increased Aβ coverage in the APP-KI model (subsection “APPPS1 and APP-KI mice show similar dynamics of amyloid plaque deposition, but differ in plaque fibrillization”), supporting that Aβ conformation/fibrillarity, rather than plaque size, are responsible for microglial changes. To better characterize different amyloid types in APPPS1 and APP-KI mouse models, we have performed spectral analysis using the two luminescent conjugated oligothiophenes (LCOs), qFTAA and hFTAA that preferably bind to dense core/fibrillar or pre-fibrillar Aβ, respectively. These new data are included into Figure 8—figure supplement 1A-D and reveal that dense core fibrillar Aβ could only be detected in the APPPS1 model at the age of 3 months. In contrast, fibrillar Aβ could be detected in both models at the age of 12 months, corresponding to the similar proteomic signatures (MARPs) and supporting the hypothesis that fibrillar Aβ may be triggering proteomic changes. Nevertheless, our data cannot exclude the effect of dystrophic neurites onto microglial populations and we therefore toned down our statement “Microglial recruitment is triggered by fibrillar Aβ and not by dystrophic neurites” into “Microglial recruitment correlates with fibrillar Aβ”.

Furthermore, according to reviewers’ recommendations, we have expanded our study and examined a correlation between microglial activation and pyroglu-Aβ species (pE3-Aβ) that are included into the new Figure 9—figure supplement 1A-D (subsection “Microglial recruitment correlates with by fibrillar Aβ”). We detected and quantified pE3-Aβreactivity in 12 month old APPPS1 and APP-KI animals (Figure 9—figure supplement 1B-D). However, we could not detect pyroglu-Aβ signal in 3 month old animals (Figure 9—figure supplement 1A), suggesting that it is less likely that this modification is responsible for microglial recruitment and differences in the proteome between the models.